# Declines in HIV incidence among men and women in a South African population-based cohort

Alain Vandormael 1,2,3,4*, Adam Akullian 5,6, Mark Siedner[1,7,8], Tulio de Oliveira[4,6,9], Till Bärnighausen[1,3,10] & Frank Tanser[1,2,9,11]

Over the past decade, there has been a massive scale-up of primary and secondary prevention services to reduce the population-wide incidence of HIV. However, the impact of these services on HIV incidence has not been demonstrated using a prospectively followed, population-based cohort from South Africa—the country with the world's highest rate of new infections. To quantify HIV incidence trends in a hyperendemic population, we tested a cohort of 22,239 uninfected participants over 92,877 person-years of observation. We report a 43% decline in the overall incidence rate between 2012 and 2017, from 4.0 to 2.3 seroconversion events per 100 person-years. Men experienced an earlier and larger incidence decline than women (59% vs. 37% reduction), which is consistent with male circumcision scale-up and higher levels of female antiretroviral therapy coverage. Additional efforts are needed to get more men onto consistent, suppressive treatment so that new HIV infections can be reduced among women.

[1] Africa Health Research Institute (AHRI), Private Bag X7, Durban 4013, South Africa. [2] School of Nursing and Public Health, University of KwaZulu-Natal (UKZN), Durban 4041, South Africa. [3] Heidelberg Institute for Global Health (HIGH), University of Heidelberg, Heidelberg 69120, Germany. [4] KwaZulu-Natal Research Innovation and Sequencing Platform (KRISP), College of Health Sciences, UKZN, Durban 4013, South Africa. [5] Institute for Disease Modeling, Bellevue, Washington 98005, USA. [6] Department of Global Health, University of Washington, Seattle, Washington 98195, USA. [7] Division of Infectious Diseases, Department of Medicine, Massachusetts General Hospital, Boston, Massachusetts 02114, USA. [8] Harvard Medical School, Boston, Massachusetts 02115, USA. [9] Centre for the AIDS Programme of Research in South Africa (CAPRISA), Durban 4013, South Africa. [10] Department of Global Health and Population, Harvard T.H. Chan School of Public Health, Boston, Massachusetts 02115, USA. [11] Lincoln Institute for Health, University of Lincoln, Lincoln LN6 7TS, UK. *email: vandormaela@ukzn.ac.za

Over the past decade, there has been a massive scale-up of treatment and prevention services to bring the HIV epidemic under control[1]. These efforts have led to an estimated 50% reduction in the number of AIDS-related deaths over the last 6 years[2]. However, progress to reduce the HIV incidence rate (IR) by 75% is off track[3], with 1.8 million new infections occurring in 2017[4]. The Joint United Nations Programme on HIV and AIDS (UNAIDS) has recently described the limited success in reducing new HIV infections, which is unmatched by the success in reducing AIDS-related deaths, as a "prevention crisis"[5].

There is an urgent need to measure the impact of primary and secondary prevention services on long-term HIV incidence trends in southern Africa—the region accounting for one-third of all new infections[4,6]. In eastern Africa, estimates of HIV incidence declines have varied between 40–50% following increased anti-retroviral therapy (ART) and voluntary medical male circumcision (VMMC) coverage[7,8]. Nevertheless, reductions in incidence have not been rigorously demonstrated using a prospectively followed, population-based cohort from southern Africa, which has a much higher rate of new HIV infections and has received substantial domestic and international investment to achieve epidemic control[1,4]. Attention is focused on the country with the world's largest treatment and prevention program: South Africa[9].

One major challenge in reliably measuring HIV incidence trends in southern Africa (as well as the broader African region) has been the limited number of population-based cohort studies. Previous estimates of incidence declines in the region have been derived from mathematical models or cross-sectional assay-based studies[5,10,11]. However, mathematical models rely on strong assumptions and non-representative data (e.g., women attending antenatal clinics), while cross-sectional studies do not track the same participants over time[12,13]. The repeated testing of a complete population of participants until HIV seroconversion is generally recognized as the gold-standard approach for measuring trends in the incidence of infection[14].

In our previous work, we demonstrated a clear relationship between ART coverage in the surrounding local community and individual risk of HIV acquisition in a hyperendemic South African setting[15]. However, we have yet to report on long-term, population-wide HIV incidence trends and their relation to the scale-up of primary and secondary prevention services. Specifically, we use one of the world's largest population-based HIV cohorts, with over 90,000 person-years of observation, to quantify sex-specific trends in HIV incidence following changes in ART coverage, prevalence of detectable viremia, condom use, and male circumcision. We leverage several differing methodologies to validate our estimates and ensure the robustness of our findings.

We report that the overall incidence of HIV infection between 2012 and 2017 declined by 43%. Our key finding is that men experienced earlier and larger declines than women, plausibly due to a sex differential in the uptake of primary and secondary prevention services. Specifically, we show that male incidence declined by 59%, from 2.5 to 1.0 seroconversion events per 100 person-years, which is consistent with VMMC scale-up in 2009 and female ART coverage surpassing 35% in 2012. We also report a 37% reduction in the female incidence between 2014 and 2017, from 4.9 to 3.1 seroconversion events per 100 person-years, which occurred after male ART coverage reached 35%. Thus, although HIV prevention efforts should continue to focus on both men and women, there is an urgent need to get more men onto consistent, suppressive ART so that new HIV infections can be further reduced among women.

## Results

**Population-based HIV testing platform.** The Africa Health Research Institute (AHRI) runs a population-based HIV testing platform in the Hlabisa sub-district of KwaZulu-Natal[16]. Approximately 90,000 persons reside in 11,000 households, which are mostly scattered across the 438 km$^2$ study area. As is typical for a rural South African setting, there are several informal peri-urban settlements and a single urban township. The majority of the residents are Zulu-speaking (black) Africans.

Two to three times yearly, trained field workers visit all households in the study area to interview a key resident informant. The key informant is often the household head or the most senior household member; if not available, other suitable household members are selected. The key informant provides information on the physical attributes of the household; the resident members and their relationship to one another; members who join, leave, or die; and the migration patterns of members, including place of origin and destination. For each death in the household, the key informant completes a detailed verbal autopsy questionnaire administered by the field-worker. To undertake annual HIV testing, field workers obtain written consent from eligible participants who are present, mentally able, and older than 15 years of age. Participants first answer questions about their sexual health, relationship history, use of condoms with a sexual partner, and circumcision status (men only between 2009 and 2016). The field workers then extract dried blood spot (DBS) samples for HIV and viral load testing[17,18].

Prevention services at the 17 public healthcare clinics in or adjacent to the study area have included HIV testing and counseling, condom distribution, ART availability, and voluntary medical circumcision for men. The local VMMC program was started in 2009 and the national HIV testing and counseling services were expanded in 2010. ART became freely available nationwide in 2004, with a CD4+ T-cell count eligibility criteria of <200 cells/µL. In 2010, eligibility was extended to pregnant woman with CD4+ T-cell counts <350 cells/µL and to patients with active tuberculosis[19]. All patients with CD4+ T-cell counts <350 cells/µL became eligible for ART in 2011. In 2015, ART was made available to HIV-positive pregnant women regardless of CD4+ T-cell count and to all late adolescents and adults with CD4+ T-cell counts <500 cells/µL. In 2016, all HIV-positive persons became eligible for ART[20].

We linked ~60% of all HIV-positive participants to the 17 public healthcare clinics in the study area. From these clinics we obtained records on ART initiation and clinic visit dates. The remaining ~40% of HIV-positive participants had either never been initiated on ART or initiated ART at other clinics outside this setting.

**Survey participation.** Table 1 shows the participation rates in the HIV survey. On average, 92% (range: 84–97%) of the enumerated participants were eligible for an HIV test, 85% (range: 76–93%) of the eligible participants were contacted at the household visit, and 41% (range: 34–52%) of the contacted participants tested for HIV. Supplementary Fig. 1 shows that the HIV testing rate did not differ substantially by age or sex over the observation period. Because of the annual surveys, the cumulative percentage of participants that had at least one HIV test increased from 60% in 2005 to 68% in 2010 and to 80% in 2017. Overall, a total of 53,167 participants tested for HIV over 158,369 person-visits. Table 1 also shows the number of eligible HIV-negative participants by year (column 9) and the number of these participants that had a repeat HIV test (column 10). On average, 76% (range: 69–87%) of the eligible HIV-negative participants entered into the HIV

**Table 1 Summary of testing participation in the HIV survey and HIV incidence cohort.**

| | HIV survey[a] | | | | | | HIV incidence Ccohort[b] | | | |
| | Eligible/enumerated[c] | | Contact/eligible[d] | | Tested/contacted[e] | | Ever test[f] | Eligible[g] | Repeat-testers[h] | |
| Year | No./total no. | % | No./total no. | % | No./total no. | % | % | No. | No. | % |
| 2005 | 25,203/ 26,477 | (95.2) | 22,105/ 25,203 | (87.7) | 9422/22,105 | (42.6) | 59.8 | 11,396 | 9936 | (87.2) |
| 2006 | 23,536/ 26,209 | (89.8) | 21,538/ 23,536 | (91.5) | 8643/21,538 | (40.1) | 61.8 | 13,973 | 11,522 | (82.5) |
| 2007 | 28,364/ 32,844 | (86.4) | 25,689/ 28,364 | (90.6) | 9957/25,689 | (38.8) | 60.4 | 16,064 | 12,898 | (80.3) |
| 2008 | 31,020/ 36,228 | (85.6) | 28,859/ 31,020 | (93.0) | 10,032/ 28,859 | (34.8) | 61.5 | 17,958 | 14,163 | (78.9) |
| 2009 | 27,077/ 32,072 | (84.4) | 24,795/ 27,077 | (91.6) | 8858/ 24,795 | (35.7) | 66.2 | 19,074 | 14,816 | (77.7) |
| 2010 | 32,173/ 34,968 | (92.0) | 26,520/ 32,173 | (82.4) | 11,228/ 26,520 | (42.3) | 68.0 | 21,483 | 16,367 | (76.2) |
| 2011 | 32,127/ 32,978 | (97.4) | 25,586/ 32,127 | (79.6) | 10,385/ 25,586 | (40.6) | 70.7 | 23,052 | 17,242 | (74.8) |
| 2012 | 30,506/ 32,327 | (94.4) | 23,145/ 30,506 | (75.9) | 7916/23,145 | (34.2) | 69.6 | 24,519 | 18,199 | (74.2) |
| 2013 | 30,768/ 32,964 | (93.3) | 24,840/ 30,768 | (80.7) | 9912/24,840 | (39.9) | 72.5 | 26,500 | 19,482 | (73.5) |
| 2014 | 31,225/ 33,013 | (94.6) | 24,471/ 31,225 | (78.4) | 9541/24,471 | (39.0) | 74.1 | 28,027 | 20,532 | (73.3) |
| 2015 | 30,509/ 32,211 | (94.7) | 27,085/ 30,509 | (88.8) | 13,096/ 27,085 | (48.4) | 78.0 | 30,143 | 21,832 | (72.4) |
| 2016 | 32,502/ 34,044 | (95.5) | 28,239/ 32,502 | (86.9) | 14,737/ 28,239 | (52.2) | 80.4 | 32,481 | 22,943 | (70.6) |
| 2017 | 31,687/ 34,542 | (91.7) | 26,240/ 31,687 | (82.8) | 11,430/ 26,240 | (43.6) | 80.5 | 33,837 | 23,445 | (69.3) |
| Average | | (91.9) | | (85.4) | | (40.9) | | | | (76.2) |

[a]Trained field workers visit all households in the surveillance area on an annual basis to take dried blood spots for HIV testing. Field workers also record data on whether participants were contacted and tested at the household visit
[b]Participants entered into the HIV incidence cohort if they had an earliest HIV-negative test result followed by at least one valid HIV test result
[c]Enumerated refers to all individuals living in the household, as identified by a key household informant at the time of the field-worker visit. Eligibility was defined as being >15 years of age, mentally able, and a resident of the household. The column shows the number and percentage of enumerated individuals that were eligible for an HIV test
[d]Shows the number and percentage of eligible participants that were successfully contacted (i.e., that were present in the household) at the time of the field-worker visit
[e]Shows the number and percentage of contacted participants that were tested for HIV
[f]Shows the cumulative proportion of participants that tested at least once for HIV since the start of HIV testing in 2003
[g]Shows the cumulative number of HIV-negative participants since 2003 that were eligible for entry into the HIV incidence cohort in 2005
[h]Shows the number of eligible HIV-negative participants that had a repeat-test, entered the HIV cohort, and contributed person-time to the incidence analysis. For example, if a participant had a first and last HIV-negative test in 2006 and 2010, then he she is included in the year 2006, 2007, 2008, 2009, and 2010, irrespective of the number of missed tests during this exposure time. The last column gives the annual percentage of eligible HIV-negative participants that entered the HIV cohort

cohort (column 11). The median length of the censoring interval was 3 years.

**Unadjusted HIV incidence rates and rate ratios.** Of the participants that tested for HIV, 22,239 had a negative test result followed by at least one valid test result. Of these repeat testers, 12,609 (57%) were women and 9630 (43%) were men. The median (interquartile range age was 22 (19–31) and 24 (19–37) years for men and women, respectively. We observed 770 seroconversion events per 38,071 person-years of follow-up among men and 2410 seroconversion events per 54,806 person-years of follow-up among women.

Tables 2 and 3 show the HIV events, person-years, unadjusted IRs, and unadjusted IR ratios (IRRs) by year. Declines in HIV incidence occurred once opposite-sex ART coverage surpassed 35% (Model 1) and ART eligibility criteria were removed in 2016 (Model 2). Between 2012 and 2017, the HIV IR (95% confidence interval [CI]) among men declined by 59%, from 2.49 (1.83–3.37) to 1.01 (0.58–1.76) seroconversion events per 100 person-years (see Fig. 1 and Table 2). HIV incidence was higher among women and relatively flat between 2005 and 2013, at around 4.3 seroconversion events per 100 person-years. Thereafter, between 2014 and 2017, the female HIV incidence declined by 37%, from

4.89 (4.09–5.84) to 3.06 (2.38–3.94) seroconversion events per 100 person-years (see Fig. 1 and Table 3). Between 2012 and 2017, the overall HIV IR among men and women declined from 3.94 (3.37–4.60) to 2.25 (1.79–2.83) events per 100 person-years, a 43% reduction (see Supplementary Table 1).

**Age-specific HIV incidence and HIV prevalence.** Figure 2 shows the HIV incidence and HIV prevalence for the 15–49, 15–29, and 30–54 years age groups by sex. Between 2005 and 2017, the HIV prevalence for the 15–54-year age group increased from 14% to 20% among men and from 25% to 41% among women. HIV incidence was highest among women aged 15–29 years, which decreased markedly between 2014 and 2017. HIV prevalence was extremely high among women aged 30–49 years, which increased from 32% in 2005 to 59% in 2017. The female HIV incidence in this older age group did not decline until the last year of the observation period.

**ART coverage and prevalence of detectable viremia.** Supplementary Tables 2 and 3 show the number and percentage of HIV-positive men and women on ART, respectively. Among HIV-positive women, ART coverage increased from 2.1% in 2005

**Table 2 Incidence rates (IRs, unadjusted) and incidence rate ratios (IRRs, unadjusted and adjusted) for men (_N_ = 9630) by female ART coverage, ART scale-up period, and year.**

|  | Events | Person-years | Inc rate (95% CI) | Unadj. IRR (95% CI) | _P_-value | Adj. IRR (95% CI)[d] | _P_-value |
|---|---|---|---|---|---|---|---|
| Model 1: By female ART coverage[a] |  |  |  |  |  |  |  |
| 0–9% | 149 | 6809 | 2.20 (1.81–2.68) | Ref. | – | Ref. | – |
| 10–24% | 339 | 14,175 | 2.40 (2.12–2.71) | 1.09 (0.85–1.39) | 0.495 | 0.72 (0.58–0.89) | <0.01 |
| 25–34% | 74 | 3221 | 2.30 (1.70–3.11) | 1.05 (0.73–1.51) | 0.805 | 0.64 (0.46–0.89) | <0.01 |
| 35–55% | 294 | 16,669 | 1.76 (1.55–2.01) | 0.80 (0.63–1.02) | 0.068 | 0.52 (0.40–0.68) | <0.01 |
| Model 2: By ART scale-up period[b] |  |  |  |  |  |  |  |
| 2005–2010 | 489 | 20,984 | 2.33 (2.12–2.57) | Ref. | – | Ref. | – |
| 2011–2015 | 315 | 15,308 | 2.06 (1.82–2.33) | 0.88 (0.75–1.04) | 0.147 | 0.75 (0.62–0.90) | <0.01 |
| 2016–2017 | 52 | 4582 | 1.15 (0.82–1.60) | 0.49 (0.34–0.70) | <0.01 | 0.52 (0.38–0.72) | <0.01 |
| Model 3: By year[c] |  |  |  |  |  |  |  |
| 2005 | 69 | 3239 | 2.14 (1.57–2.93) | 0.86 (0.56–1.33) | 0.504 | 1.50 (1.04–2.16) | 0.028 |
| 2006 | 80 | 3569 | 2.24 (1.69–2.96) | 0.90 (0.59–1.36) | 0.621 | 1.46 (1.02–2.08) | 0.038 |
| 2007 | 84 | 3634 | 2.30 (1.74–3.05) | 0.93 (0.61–1.42) | 0.724 | 1.06 (0.74–1.52) | 0.755 |
| 2008 | 86 | 3679 | 2.35 (1.78–3.09) | 0.94 (0.63–1.42) | 0.783 | 0.98 (0.68–1.41) | 0.910 |
| 2009 | 85 | 3483 | 2.45 (1.85–3.24) | 0.99 (0.65–1.50) | 0.947 | 1.10 (0.78–1.55) | 0.599 |
| 2010 | 83 | 3377 | 2.45 (1.85–3.25) | 0.99 (0.65–1.50) | 0.952 | 1.06 (0.76–1.48) | 0.732 |
| 2011 | 74 | 3221 | 2.30 (1.70–3.11) | 0.93 (0.58–1.47) | 0.744 | 0.92 (0.65–1.30) | 0.621 |
| 2012 | 75 | 3039 | 2.49 (1.83–3.37) | Ref. | – | Ref. | – |
| 2013 | 67 | 3033 | 2.22 (1.64–3.01) | 0.89 (0.57–1.39) | 0.620 | 0.92 (0.65–1.30) | 0.621 |
| 2014 | 56 | 3046 | 1.83 (1.29–2.59) | 0.73 (0.45–1.19) | 0.210 | 0.65 (0.44–0.96) | 0.029 |
| 2015 | 41 | 2967 | 1.39 (0.94–2.07) | 0.56 (0.34–0.93) | 0.025 | 0.65 (0.44–0.95) | 0.027 |
| 2016 | 32 | 2602 | 1.24 (0.79–1.95) | 0.50 (0.29–0.87) | 0.014 | 0.70 (0.47–1.04) | 0.074 |
| 2017 | 20 | 1979 | 1.01 (0.58–1.76) | 0.41 (0.22–0.76) | <0.01 | 0.39 (0.23–0.67) | <0.01 |

HIV events, person-years, unadjusted IRs, and unadjusted adjusted IRRs are averages over 300 datasets generated by imputing a single random infection date within the censoring interval. Rubin's rules were used to calculate 95% confidence intervals
[a]Opposite-sex ART coverage used because of the generalized, heterosexual HIV epidemic in South Africa, with 0–9% as the reference category
[b]Time intervals were defined by changes in national criteria for ART eligibility, with 2005–2010 as the reference period
[c]Time intervals were defined by year, with 2012 as the reference year
[d]Estimates adjusted for age, self-reported condom use, circumcision and marital status, household assets index, cumulative time spent outside surveillance area, and opposite-sex HIV prevalence in the surrounding community (see Supplementary Table 4 for full results)

**Table 3 Incidence rates (IRs, unadjusted) and incidence rate ratios (IRRs, unadjusted and adjusted) for women (_N_ = 12,609) by male ART coverage, ART scale-up period, and year.**

|  | Events | Person-years | Inc rate (95% CI) | Unadj. IRR (95% CI) | _P_-value | Adj. IRR (95% CI)[d] | _P_-value |
|---|---|---|---|---|---|---|---|
| Model 1: By male ART coverage[a] |  |  |  |  |  |  |  |
| 0–9% | 642 | 14,653 | 4.38 (4.00–4.79) | Ref. | – | Ref. | – |
| 10–24% | 674 | 14,549 | 4.63 (4.24–5.07) | 1.06 (0.92–1.21) | 0.422 | 0.99 (0.89–1.11) | 0.891 |
| 25–34% | 847 | 17,570 | 4.82 (4.47–5.21) | 1.10 (0.98–1.24) | 0.114 | 0.87 (0.77–0.97) | 0.016 |
| 35–55% | 416 | 11,014 | 3.78 (3.38–4.23) | 0.86 (0.75–1.00) | 0.043 | 0.68 (0.59–0.78) | <0.01 |
| Model 2: By ART scale-up period[b] |  |  |  |  |  |  |  |
| 2005–2010 | 1,316 | 29,203 | 4.51 (4.25–4.78) | Ref. | – | Ref. | – |
| 2011–2015 | 1,030 | 21,796 | 4.73 (4.42–5.05) | 1.05 (0.96–1.15) | 0.317 | 0.85 (0.77–0.93) | <0.01 |
| 2016–2017 | 234 | 6789 | 3.45 (2.95–4.02) | 0.76 (0.65–0.90) | <0.01 | 0.63 (0.55–0.74) | <0.01 |
| Model 3: By year[c] |  |  |  |  |  |  |  |
| 2005 | 186 | 4567 | 4.08 (3.40–4.90) | 0.84 (0.65–1.08) | 0.166 | 1.11 (0.90–1.37) | 0.339 |
| 2006 | 223 | 5008 | 4.45 (3.77–5.27) | 0.91 (0.71–1.17) | 0.465 | 1.16 (0.95–1.42) | 0.157 |
| 2007 | 231 | 5078 | 4.56 (3.86–5.39) | 0.93 (0.73–1.19) | 0.585 | 1.18 (0.97–1.44) | 0.097 |
| 2008 | 231 | 5050 | 4.58 (3.89–5.40) | 0.94 (0.74–1.19) | 0.604 | 1.13 (0.92–1.38) | 0.234 |
| 2009 | 220 | 4818 | 4.58 (3.85–5.44) | 0.94 (0.73–1.20) | 0.604 | 1.14 (0.94–1.39) | 0.176 |
| 2010 | 221 | 4681 | 4.72 (3.98–5.61) | 0.97 (0.75–1.25) | 0.795 | 1.15 (0.94–1.40) | 0.169 |
| 2011 | 209 | 4566 | 4.59 (3.85–5.47) | 0.94 (0.73–1.21) | 0.624 | 0.97 (0.79–1.20) | 0.803 |
| 2012 | 214 | 4319 | 4.95 (4.14–5.92) | 1.01 (0.78–1.31) | 0.920 | 1.02 (0.83–1.25) | 0.846 |
| 2013 | 211 | 4354 | 4.85 (4.05–5.81) | 0.99 (0.76–1.30) | 0.961 | 1.00 (0.82–1.21) | 0.968 |
| 2014 | 211 | 4329 | 4.89 (4.09–5.84) | Ref. | – | Ref. | – |
| 2015 | 182 | 4225 | 4.31 (3.58–5.20) | 0.88 (0.67–1.16) | 0.368 | 0.86 (0.71–1.06) | 0.164 |
| 2016 | 141 | 3783 | 3.74 (3.04–4.61) | 0.77 (0.58–1.02) | 0.064 | 0.78 (0.63–0.97) | 0.026 |
| 2017 | 92 | 3005 | 3.06 (2.38–3.94) | 0.63 (0.46–0.86) | <0.01 | 0.65 (0.51–0.83) | <0.01 |

HIV events, person-years, unadjusted IRs, and unadjusted adjusted IRRs are averages over 300 datasets generated by imputing a single random infection date within the censoring interval. Rubin's rules were used to calculate 95% confidence intervals
[a]Opposite-sex ART coverage used because of the generalized, heterosexual HIV epidemic in South Africa, with 0–9% as the reference category
[b]Time intervals were defined by changes in national criteria for ART eligibility, with 2005–2010 as the reference period
[c]Time intervals were defined by year, with 2012 as the reference year
[d]Estimates adjusted for age, self-reported condom use, circumcision and marital status, household assets index, cumulative time spent outside surveillance area, and opposite-sex HIV prevalence in the surrounding community (see Supplementary Table 5 for full results)

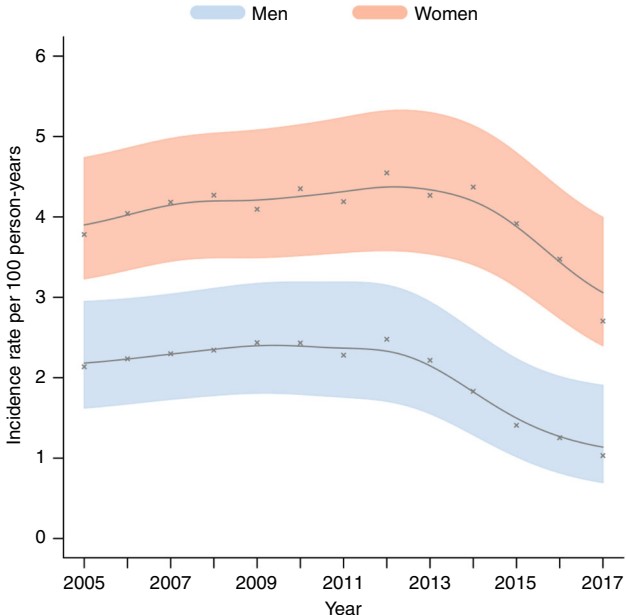

**Fig. 1** HIV incidence rates with 95% confidence intervals (CIs). The figure shows that male and female HIV incidence declined after 2012 and 2014, respectively, with an overall decline of 43% between 2012 and 2017.

to 24.6% in 2010, and to 50.6% in 2017. ART coverage was lower among HIV-positive men, which increased from 1.5% in 2005 to 21.4% in 2010, and then to 38.4% in 2017. As reported elsewhere[21], the population prevalence of detectable viremia among women declined from 72.8% in 2011 to 62.0% in 2013 and 55.3% in 2014. Among men, it declined from 77.8% in 2011 to 70.4% in 2013 and 67.2% in 2014 (see Fig. 3).

**Condom use and circumcision status.** Supplementary Tables 2 and 3 also show the number and percentage of men and women reporting condom use in relationships. As reported by women, condom use among male partners increased from 33.8% in 2005 to 60.5% in 2011, and averaged around 64% thereafter. Among men, self-reported condom use was stable between 2011 and 2017, at around 70%, having risen from 43.6% in 2005 to 71.6% in 2011. Supplementary Table 2 shows that self-reported circumcision coverage among men in the study area increased from 3.0% to 32.9% between 2009 and 2016. Among circumcised men, the HIV IR declined by 59%, from 1.24 (0.57–2.69) to 0.5 (0.16–1.57) events per 100 person-years between 2012 and 2016 (Fig. 4). During the same period, the IR among uncircumcised men declined from 3.01 (2.16–4.18) to 1.73 (1.01–2.97) events per 100 person-years—a comparatively lower reduction of 42%.

**Adjusted HIV incidence rate ratios.** We show the adjusted IRRs by ART coverage category, ART eligibility period, and year in Tables 2 and 3, and the full results in Supplementary Tables 4 and 5. For Model 1, declines in the adjusted male and female IRRs were associated with increased opposite-sex ART coverage, holding HIV prevalence and other key risk factors for HIV acquisition constant. Declines in the adjusted male and female IRRs were also significantly associated with more inclusive ART eligibility criteria, as shown in Model 2. Model 3 confirms declines in the adjusted male and female IRRs after 2012 and 2014, respectively.

Results show that men in the 25–29 age group and women in the 20–24 age group were at the greatest risk of HIV

infection, when compared with 15–19-year-olds (Supplementary Tables 4 and 5). Circumcised men had a significantly lower adjusted IRR when compared with uncircumcised men (Supplementary Table 4). Compared with women, uncircumcised and circumcised men had a significantly lower adjusted IRR (Supplementary Table 6). Greater time spent outside of the study area and higher levels of HIV prevalence were associated with an increased risk of HIV acquisition across the multivariate models (Supplementary Tables 4–6).

**Trends in measures from 2010 to 2017.** In Fig. 3 we present trends in the sex-specific HIV incidence, self-reported condom use, self-reported male circumcision, opposite-sex ART coverage, and opposite-sex prevalence of detectable viremia. The left panel shows the decline in male incidence coincided with increased circumcision coverage, female ART coverage surpassing 35% after 2012, and a decline in the female prevalence of detectable viremia. The right panel shows the female incidence declined once male ART coverage surpassed 35% after 2014, with a lower reduction in the male prevalence of detectable viremia when compared with women. Condom use among men and women remained unchanged between 2012 and 2017.

**Discussion**

Using a prospectively followed, population-based cohort from a hyperendemic South African setting, we demonstrate a 43% decline in the incidence of HIV infection between 2012 and 2017. Men experienced an earlier and larger decline in incidence, which we attribute to the introduction of a local VMMC program in 2009, the scale-up of national testing and counseling services in 2010, and female ART coverage surpassing 35% in 2012. Among women, the comparatively lower decline in incidence began after 2014, once male ART coverage reached 35%. The unequal declines in HIV incidence are broadly consistent with a sex differential in the uptake of primary and secondary prevention services in the study area[22].

Our findings are biologically plausible: we would expect to observe earlier and larger population-level reductions in the male HIV IR following earlier VMMC scale-up and higher levels of ART coverage among women. VMMC is beneficial for men and has been associated with a reduced risk of male HIV acquisition in sub-Saharan African settings[23–25]. In 2009, a local VMMC program was introduced into the study area and by 2016 self-reported circumcision coverage had reached 33%. Our results confirm the preventative benefits of circumcision for men, who had a lower incidence of infection when compared with uncircumcised men. Uncircumcised male participants also had a markedly lower rate of new HIV infections than women, which is likely to have contributed to the earlier and larger declines in the overall incidence among men.

Men also likely benefited from higher female engagement with the healthcare system. At least partly due to perinatal HIV screening and treatment programs, women are more likely than men to test for HIV, initiate ART early, and achieve long-term undetectable viremia[26–28]. Given the generalized, heterosexual HIV epidemic in sub-Saharan Africa, this means that men will have a comparatively lower risk of meeting female sexual partners with detectable HIV viremia and acquiring HIV (on average and under the assumption of a well-mixed population sexual network)[21,29,30]. We have previously found that the prevalence of detectable viremia, when combined with the underlying spatial variation in the proportion of the population infected (HIV prevalence), was strongly associated with future HIV infections. Specifically, every 1% increase in the overall proportion of the population having detectable viremia was independently

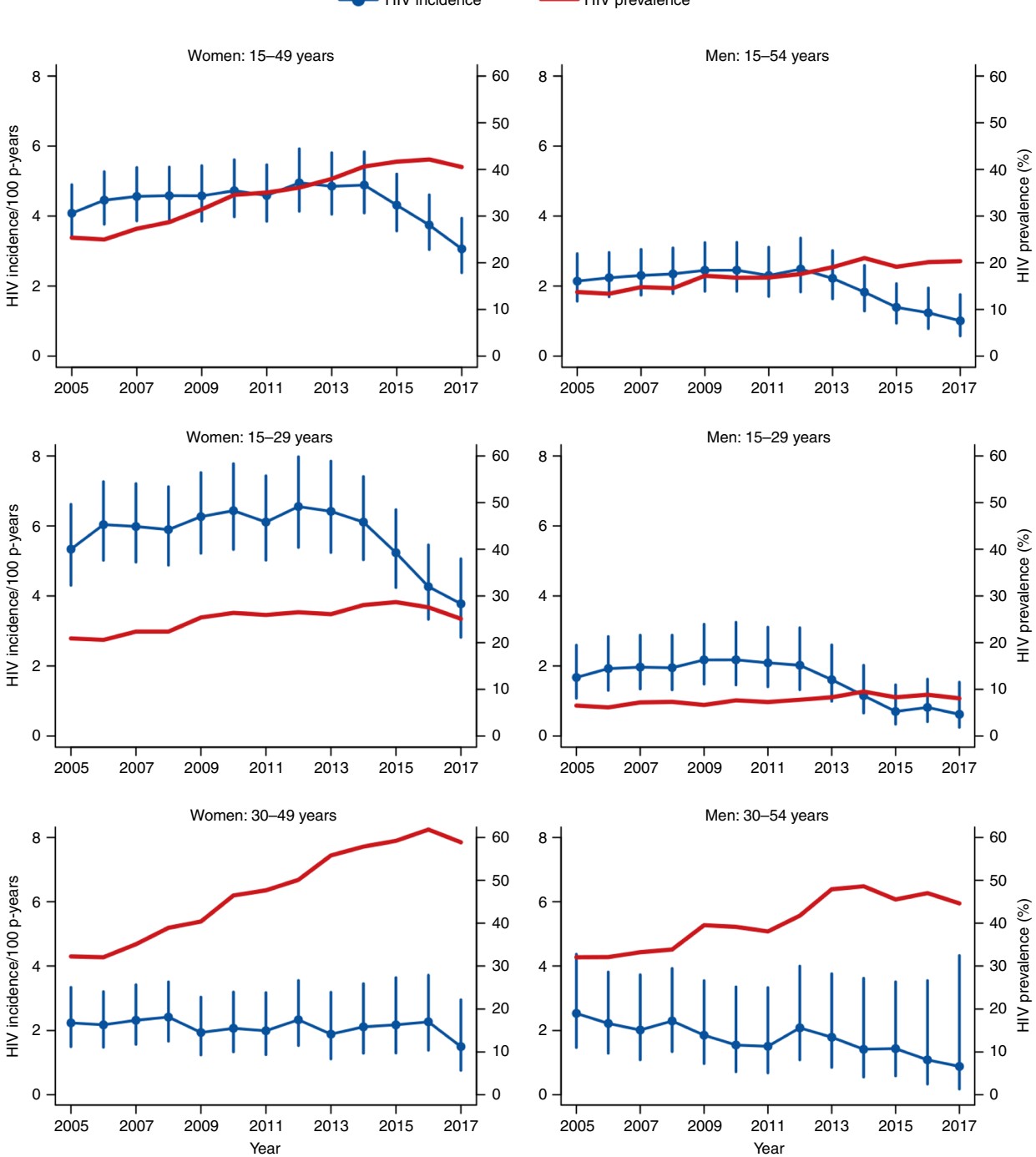

**Fig. 2** HIV incidence rates (with 95% CIs) and HIV prevalence by sex and age. The figure shows that HIV incidence was highest among younger participants (15–29 years) and HIV prevalence was highest among older women (30–49 years). The largest declines in HIV incidence occurred among younger women, whereas HIV incidence was relatively flat among older women.

associated with a 6.3% increase in the prospective risk of HIV acquisition[29]. We hypothesize that higher levels of ART coverage and hence lower population levels of detectable viremia among women were two important factors that contributed to larger and earlier declines in HIV incidence among men.

Our finding of a large HIV incidence decline is consistent with other regions in sub-Saharan Africa. Recently, an ongoing cohort study from the Rakai district in Uganda estimated a 42% decline in the overall HIV IR between 2012 and 2016. This decline was attributed to increased ART coverage, VMMC scale-up, and possibly reduced sexual activity in late adolescence[7]. In western Kenya, the scale-up of VMMC services, and to a lesser extent increased ART coverage, was associated with a 50% reduction in the HIV IR between 2011 and 2016[8]. Representative cross-sectional studies from eSwatini (Swaziland) and South Africa have recently reported similar reductions in HIV incidence over the last six years[31,32]. Consistent with these results, we show for the first time, in a southern African setting, significant declines in

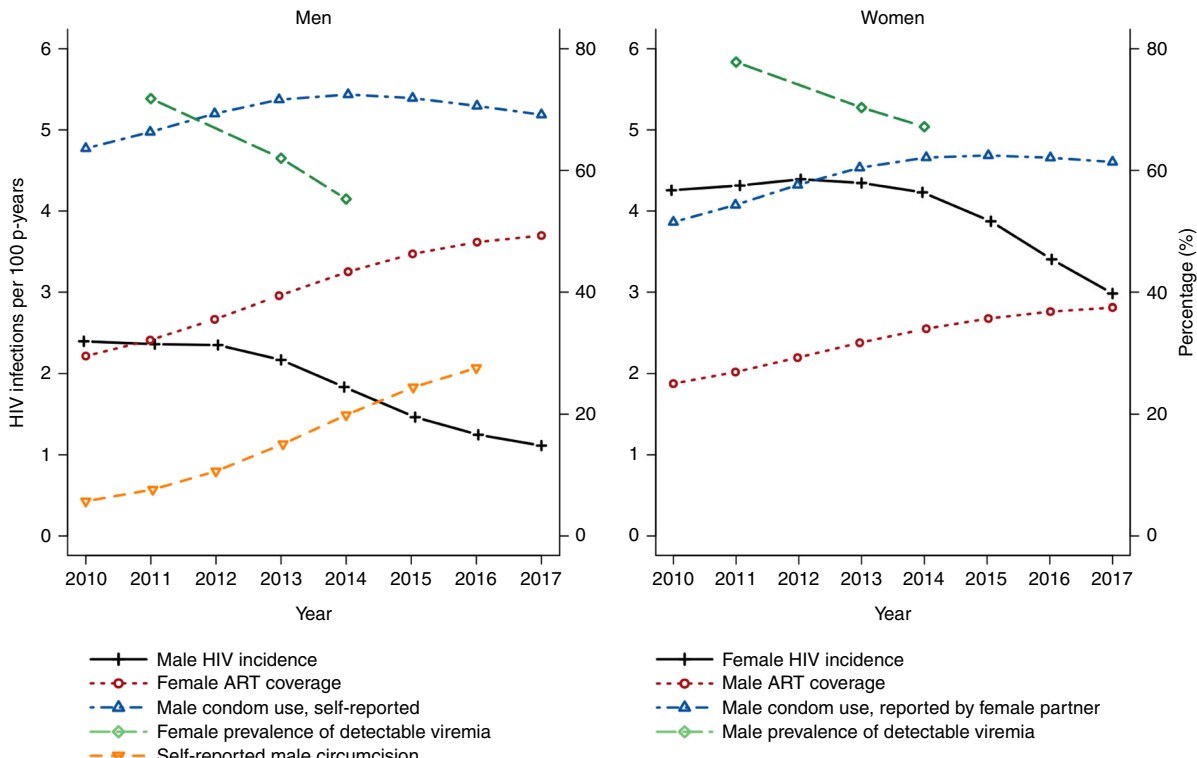

**Fig. 3** Trends in HIV incidence, self-reported condom use, self-reported male circumcision, opposite-sex ART coverage, and opposite-sex prevalence of detectable viremia. The figure shows that male HIV incidence began to decline after 2012, following increased VMMC coverage, female ART coverage surpassing 35%, and a decrease in the female prevalence of detectable viremia. Declines in female HIV incidence after 2014 correspond with male ART coverage reaching 35% and declines in the male prevalence of detectable viremia.

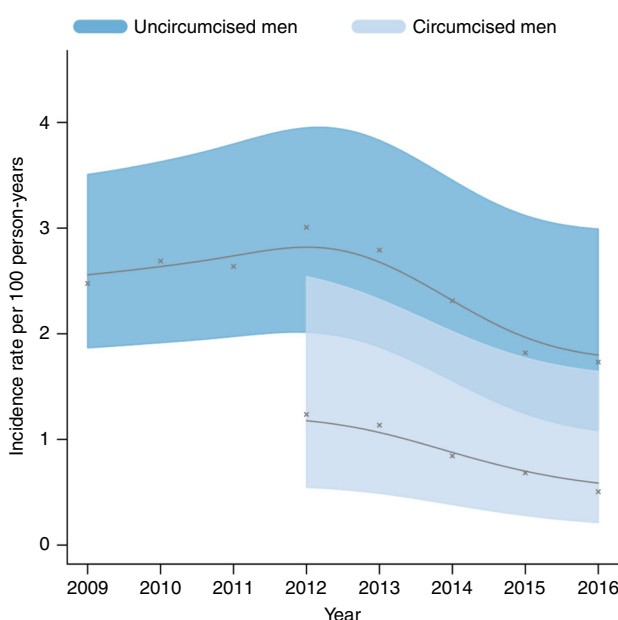

**Fig. 4** Age-adjusted HIV incidence rates for men who reported being uncircumcised (N = 5134) and circumcised (N = 2306). Due to a small number of recorded seroconversions between 2009 and 2011, the incidence rate for circumcised men is plotted from 2012 to 2016. The figure shows that uncircumcised men had a lower incidence of HIV infection, and that declines in HIV incidence occurred after 2012 for both uncircumcised and circumcised men.

the HIV IR using data from a prospectively followed, population-based cohort study.

The strength of our study is that we followed a cohort of 22,239 uninfected men and women to identify new HIV infections over time, which is the gold-standard approach for IR estimation[14]. We collected detailed demographic data on age, sex, relationship status, migration history, condom use, and household socio-economic status, and calculated the HIV prevalence of the participant's surrounding community. To determine ART status, we linked HIV-positive participants to their clinic records at public healthcare facilities in or adjacent to the study area. Further, we obtained viral load measurements from all HIV-positive blood samples in 2011, 2013, and 2014 to quantify the population prevalence of detectable viremia. We therefore had a unique opportunity to measure trends in the incidence of HIV infection while adjusting for well-known covariates of HIV infection risk.

We undertook several sensitivity analyses to confirm the robustness of our findings. To be sure that our results were not partially explained by model selection, we obtained unadjusted IRRs by ART coverage (Model 1), ART scale-up period (Model 2), and year (Model 3). Results show similarities between the unadjusted IRRs (Tables 2 and 3) and adjusted IRRs (Supplementary Tables 4 and 5). We note that the unadjusted IRRs declined once ART coverage surpassed 35%, suggesting that treatment uptake had to reach a population threshold before reductions in HIV incidence could be observed. After adjusting for HIV prevalence and other well-established risk factors, we found that increased ART coverage was associated with steady declines in the HIV acquisition risk. In earlier analyses, we demonstrated that reductions in the HIV acquisition risk were associated with increased ART coverage at the community,

household, and serodiscordant couple levels, holding HIV prevalence and other well-established risk factors constant[15,33,34]. Given the correspondence between the unadjusted and adjusted IRRs, and between our results and previous research, it is unlikely that the observed declines in HIV incidence could be explained by model selection.

We further assessed whether missed test dates could have biased our HIV IR results. To estimate the IR, we identified uninfected participants with an earliest negative test result followed by at least one test result. We therefore considered (1) the percentage of eligible HIV-negative participants that entered into the HIV cohort and (2) the average length of the censoring interval, defined as the time between the latest HIV-negative and earliest HIV-positive test dates. First, we show that on average 76% of HIV-negative participants entered into the HIV cohort, which is a relatively high inclusion rate for a prospective population-based cohort study. Second, we restricted our analysis to repeat testers who missed no more than two consecutive test dates within the censoring interval. Our results show similar IRs for the restricted and full cohort of repeat testers (Supplementary Table 7). This is because the single random-point method, when coupled with standard multiple imputation procedures, produces robust IR estimates—even with annual testing rates as low as 40% or average censored intervals as wide as four years. We have demonstrated this result in recent work[35]. Importantly, we also show that the demographic composition (age and sex) of the HIV-negative testers and the HIV cohort remained stable throughout the observation period (Supplementary Figs. 2 and 3). Finally, we also included inverse probability weighting in the regression models to mitigate the effect of participant selection, non-testing, and dropout. These sensitivity analyses, together with the single random-point approach for IR estimation, strongly support the robustness of our study findings.

One limitation of our study is that social desirability bias may have influenced self-reported circumcision status and condom use. However, it is not clear whether this bias could explain large increases in circumcision while condom use remained relatively flat from 2012 onward. Another potential limitation is that ~40% of all HIV-positive participants were not linked to their ART records at public healthcare facilities in or adjacent to the study area. A large proportion of these patients would be ART naïve, but some may have initiated ART at other clinics outside this setting, resulting in an underestimate of ART coverage. However, underestimated ART coverage would bias our Model 1 results toward the null hypothesis (i.e., no change in the HIV IR), as we show that increased ART is associated with a reduction in HIV acquisition risk. Further, the reductions in HIV incidence by ART coverage are also consistent with reductions in HIV incidence by ART scale-up period (Model 2) and by year (Model 3). Given similar findings from other regions in sub-Saharan Africa, it is likely that the observed declines in HIV incidence are related to the scale-up of ART and other prevention services from 2009 onward.

In summary, we provide robust evidence of HIV incidence declines among men and women living in a rural and hyperendemic South African community. Our results show that the decline in HIV incidence was larger among men and occurred earlier, which is consistent with the introduction of a local VMMC program in 2009 and female ART coverage surpassing 35% in 2012[22]. Declines in the HIV incidence among women lagged behind men by 2 years and began to decline once male ART coverage reached 35% in 2014. Overall, HIV incidence remains high, with women experiencing a higher risk of HIV acquisition relative to men. We did not observe increases in condom use between 2012 and 2017. Additional efforts are needed to increase condom uptake, improve male engagement with

the healthcare system, and ensure the scale-up of primary prevention services (such as pre-exposure prophylaxis) to bring the HIV epidemic under control by 2030.

## Methods

**HIV and viral load testing**. DBS samples were extracted by field workers according to the UNAIDS and WHO Guidelines for Using HIV Testing Technologies in Surveillance[17]. The DBS samples were transported to the AHRI laboratory in Durban where HIV status was determined by antibody testing with a broad-based HIV-1/HIV-2 ELISA (Vironostika HIV-1 Microelisa System: Biomérieux, Durham, NC, USA) followed by a second ELISA (Wellcozyme HIV-1 + 2 GACELISA: Murex Diagnostics Benelux B.V., Breukelen, Netherlands). The AHRI laboratory has used the same HIV testing algorithm since 2005.

From the HIV surveys of 2011, 2013, and 2014, we obtained viral load measurements from all the HIV-positive DBS samples[21]. Nucleic acid was extracted from the DBS samples with NucliSENS EasyMag (Bordeaux, France) and a Generic HIV Charge Virale (Biocentric, Bandol, France) test was used to quantify the viral load levels. The quantification method has a lower detection limit of 1550 copies/mL[18], which we defined as the threshold for detectable viremia.

**Statistical analysis**. We measured trends in the incidence of HIV infection using a prospectively followed cohort of repeat testers. To be included in the incidence cohort, participants had to have an HIV-negative test result followed by at least one valid HIV test result[14]. We identified all repeat testers who converted from an HIV-negative to an HIV-positive result during the observation period. Assuming a uniform distribution, we imputed a single random infection date between the latest HIV-negative and earliest HIV-positive dates (the censoring interval). We then right censored the data at the latest HIV-negative date (if uninfected) or at the imputed date (if infected). The rationale for the single random-point method is provided elsewhere[35]. We calculated the IRs per 100 person-years and estimated the IR ratios with Poisson regression models. To quantify the uncertainty of our imputation procedure, we generated 300 imputed datasets and used Rubin's rules to obtain the estimates and 95% CIs[36].

We calculated trends in the HIV IR separately for men aged 15–54 years and women aged 15–49 years between 2005 and 2017. These age-specific ranges were used to ensure consistency with previous AHRI analyses[15,33,37]. We also identified pre-circumcision exposure time for men who never reported being circumcised and post-circumcision exposure time for men who reported being circumcised. We plotted trends in the HIV IR for uncircumcised men between 2009 and 2016. Because of the small number of seroconversions recorded between 2009 and 2011, we plotted the HIV IR for circumcised men between 2012 and 2016.

We calculated the unadjusted IRs and unadjusted IRRs by ART coverage (Model 1), ART scale-up period (Model 2), and year (Model 3). For Model 1, we defined ART coverage categories of 0–9% (reference), 10–24%, 25–34%, and 35–55%. For Model 2, we selected the time intervals 2005–2010 (reference), 2011–2015, and 2016–2017 based on changes to national ART eligibility criteria. For Model 3, we selected 2012 and 2014 (based on peak HIV incidence) as the reference years for men and women, respectively.

To address potential sources of confounding, we obtained adjusted IRRs using Poisson regression models. Based on previous research undertaken in the AHRI study area, we included age[33], condom use[38], marital status[39], male circumcision status[39], household socio-economic status (in tertiles)[40], cumulative time spent outside the study area[37,41], and the HIV prevalence of the participant's surrounding community[29]. We ran the analyses separately for men and women, with opposite-sex ART coverage and opposite-sex HIV prevalence covariates, and excluded male circumcision status from the female-only models.

To adjust for participant selection and dropout, we included inverse probability of survey weights in the multivariable Poisson regression models[42]. Following the approach of another study[7], we used a logistic regression model to predict the probability of a follow-up test visit given a previous test visit, adjusting for age, HIV prevalence, and time spent outside the study area. We used the same model to predict the probability of dropout. After multiplying the follow-up and dropout weights, we included the resultant weight in the multivariable model.

We calculated ART coverage as the percentage of HIV-positive participants on treatment, VMMC coverage as the percentage of male participants that reported being circumcised, and the population prevalence of detectable viremia as the percentage of HIV-positive participants with detectable viral loads. For men and women, we plotted trends in self-reported condom use, self-reported circumcision coverage (men only), opposite-sex ART coverage, and opposite-sex prevalence of detectable viremia. We undertook all analyses in R version 3.6.1.

**Ethics Approval**. Written informed consent was obtained from all participants prior to the household interview and the extraction of DBS for HIV and viral load testing. Ethics approval for data collection and use was obtained from the biomedical and ethics committee (BREC) of the University of KwaZulu-Natal (Durban, South Africa), BREC approval number BE290/16.

**Reporting summary**. Further information on research design is available in the Nature Research Reporting Summary linked to this article.

## Data availability

All relevant data supporting the key findings of this study are available within the article and its Supplementary Information files. The datasets used for the analysis presented in this study are available from the Africa Health Research Institute (AHRI) data repository https://data.africacentre.ac.za/index.php/auth/login/?destination=. To access the licensed datasets, the applicant must agree to the terms and conditions of use by completing an Application for Access to a Licensed Dataset. This request will be reviewed by the AHRI Data Release Committee, who may decide to approve the request, to deny access to the data, or to request additional information from the applicant.

## Code availability

The R code used to generate the results is available from the corresponding author upon request.

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

## Acknowledgements

This work was supported by two National Institute of Health (NIH) grants (R01HD084233 and R01AI124389). Funding for the Africa Health Research Institute's

Demographic Surveillance Information System and Population-based HIV Survey was received from the Wellcome Trust 201433/Z/16/Z. T.B. was supported by the Alexander von Humboldt Foundation through the endowed Alexander von Humboldt Professorship funded by the German Federal Ministry of Education and Research, as well as by the Wellcome Trust, the European Commission, the Clinton Health Access Initiative, and the National Institutes of Health's Fogarty International Center (D43-TW009775).

## Author contributions

A.V. and F.T. conceived the study. A.V. wrote the paper and performed the statistical analysis with oversight from F.T. A.A., M.S., T.O., T.B., and F.T. provided comments and feedback. All authors reviewed and approved the statistical analysis and final version of the manuscript.

## Competing interests

The authors declare no competing interests.
