## [Peer Review File · Nature Communications]

Reviewers' comments:

Reviewer #1 (Remarks to the Author):

Remarks to the authors:

Given the relatively intractable nature of the HIV epidemic in SA, particularly in KZN, data showing positive effects of the ongoing roll out of combined prevention measures would represent an important and encouraging finding. Data presented here are from a large population based HIV surveillance cohort with over a decade of follow-up, and such cohorts represent the gold standard for assessing trends in HIV rates. However, there are a number of issues which complicate the interpretation of the findings, of which the principal ones are:

Given the changes in the rates of HIV testing over time, there is the potential for substantial selection bias. Information is needed on how the cohort's composition in each component (enumerated, eligible, contacted, tested, followed-up) has changed over time.

For all three models in Table 2, the declines in the adjusted incidence rate ratios are substantially more pronounced than one would expect from the actual incidence rates. For example, in Model 1, observed incidence declined from 3.48 in 2005-2007 to 3.35 in 2013-2017 (a very small change) but the adjIRR dropped sharply to 0.64. This suggests there could be substantial confounding. The authors should also present the unadjusted IRRs and include more information on how their adjustments were carried out and on the most likely sources of potential confounding.

The following remarks refer to specific sections in the order in which they appear:

Authorship list:

It is surprising that the study team did not have an African collaborator (non-foreign, non-expatriate) who was sufficiently involved in study design, data collection, analyses and/or interpretation to be included among the authorship.

Abstract: The referent year used in the tables and supplementary materials is 2006, not 2012, and I suggest that the incidence observed in that year also be used in the abstract (rather than 2012).

Page 3: "... the lack of population-based cohort studies..." and "previous estimates ...have been derived from mathematical models or cross-sectional assay-based studies" suggests there are no other such cohorts in Africa. Later on, the authors themselves cite a number of such studies. The term "... only a few other population-based cohort studies...." would be more appropriate.

Page 4: HIV survey methods. The referenced paper (Ref 16, Int J Epi 2008) describing the survey reports information up to 2006/7 (12 years ago). Have the survey and outreach methods changed since then? For example, has HIV testing outreach been intensified, such that the characteristics of those being tested have changed? Reference 16 reports that the consent rate for HIV testing in 2005 and 2006 was only ~40%; the authors of this 2008 paper raise concerns about selection bias, and indicate that changes were being made both operationally and analytically to address this issue. This further underlines the need for detailed information on the actual composition of the cohort, by key characteristics, for each survey year, providing the reader with a sense of potential selection bias over time.

Has any validation been carried out on the quality of the linkage of public health clinic data to AHRI data (missing data), and on the accuracy of the clinic data (the quality of clinical data collection and recording)?

Why is the age range for men and women different?

Page 5:

Are participants included if they have missed one or more surveys, and if so, how many such skips are allowed? What proportion of persons have no, one, two or more missing time points? Imputation would suggest that the timing of incidence is random within a given time frame, but this assumption might not hold true over long periods of time, if there are significant secular changes (for example hypothetically, a large in-migration event which brings in a substantial group of high risk individuals in a given year). Supplementary data on observed HIV incidence rates allowing for only one or two missed intervals would be useful.

Does the interview collect information on numbers and types of partners since the last survey?

Page 6 (Results):

What were the criteria that rendered 23% of those enumerated ineligible for HIV testing? The proportion of persons contacted and consenting to be tested increased from 38% to 75% over time: key characteristics of persons actually tested need to be provided for each time period.

Page 7 and Figure 4:

The text and figure contradict each other. The text reports that HIV incidence among circumcised men fell from 2.2 to 0.03 per 100 py: This is probably a typo, since the figure shows the decline to achieve $\sim 0.3/100$ py. However, the footnote in the figure says HIV incidence among circumcised men fell by 99%, which based on reports by others seems very optimistic (the actual figure suggests the reduction was in the order of $\sim 85\%$).

It is puzzling that a steady decline in incidence among circumcised men began in 2009 whereas in the uncircumcised, incidence actually went up slightly between 2009 and 2013. A circumcised male experiences a fairly constant $\sim 50\text{-}60\%$ reduction in risk of HIV. Presumably, the additional decline in incidence over time in circumcised men occurs because of the protective effects of increasing rates of female partner ART use. One would expect the uncircumcised men to also show some declines in incidence between 2009-2013, as more women went on ART. It would be good to present data on circumcised and uncircumcised male characteristics over time, in order to explore if selection bias may have affected the results.

Discussion:

Page 8:

Since ART first became available in 2004, and expanded over time, the comparison should be between that 2004 and 2017. Selecting the year with the highest observed incidence rate (2012) may overstate the effects of combined HIV prevention in this population.

Tables:

The supplementary materials should include a detailed table indicated key characteristics of the enumerated, contacted and tested population for each study year, as well as characteristics of those followed up.

Table 1:

- Please define what is meant by “eligible for testing” (who is not eligible?)
- If I understand correctly, the “ever tested” rate includes anyone who has ever had an HIV test since 2005. It would be informative to know what proportion of previously HIV-neg participants received HIV re-testing in each year. How did the characteristics of those tested in each year differ over time?
- Are “ever tested” rates based on self-report, AHRI survey data, or from clinic data? If it is the latter, has the quality of the clinic linkage and clinic data been verified?
- The legend refers to “contacted from eligible” but the footnote defines this as “consented to an HIV test”. If it is the latter, why are the proportions “ever tested” so much lower than those consented?
- Are participants allowed one or more “skipped” follow-up survey visits in the calculation of incidence rates?
- Why is the 2016 population of individuals enumerated, eligible and contacted substantially lower than in prior years? I understand that the 2017/2018 data are incomplete, but 2016 should be “in the can”.
- A flow diagram could be very useful to clearly show who drops out where in the surveillance “cascade” (from enumerated to receipt of an HIV re-test) and how that has changed over time.

Table 2: incidence rate trends and IRRs by ART coverage...

- As I commented in the introduction, for all three models in Table 2, the declines in the adjusted incidence rate ratios are substantially more pronounced than one would expect from the actual incidence rates. This suggests there could be substantial confounding. The authors should also present the unadjusted IRRs and include more information on potential confounders.
- In addition, the ART coverage threshold by 2010-2012 was already 20-39%, compared to prior time intervals with much lower coverage, yet the observed incidence rate in 2010-2012 was higher than in any other period (even as one would expect the rate to start edging down, given the expanded ART coverage). What might account for this: secular factors, differences in the characteristics of those who provided samples for HIV testing, other?

S1: I

-It would be more intuitive to show uncircumcised men as the referent group: this would be consistent with how the VMMC randomized trials presented their data (i.e., would more readily indicate the degree to which VMMC is protective in the KZN population cohort).

S1-S3:

- Please specify whether the reported rate ratios are adjusted or unadjusted.

S1 and S2:

- Why do the numbers of person-years progressively decline over time? In 2017, the reported persons years represent only 45% of those in 2016: this implies a lower follow-up rate compared to the 63.4% reported in Table 1. To reiterate a prior question, are participants allowed skips in follow-up and if so, how many? Is there a difference in HIV incidence between those followed in consecutive surveys compared to those who skip rounds?

- It is intriguing that the communities with the highest cumulative out-migration rate would also have the highest incidence. Are these highly mobile communities overall (did they also have the highest in-migration rates, potentially receiving more individuals who are not yet accessing prevention and treatment services?)

S6:

- Although the legend refers to the ART coverage threshold, it would seem that only the first section of the table is related to actual ART coverage, while the incidence rates by other characteristics are for all time intervals combined.

Reviewer #2 (Remarks to the Author):

The paper describes a 45% decline over time, between 2005-2015, in HIV incidence in the Hlabisa sub-district of KwaZulu-Natal region of South Africa, based on an open, longitudinal, population based cohort that has undergone approximately annual rounds of HIV testing. The methods and analytic approach are generally very rigorous, and manuscript is very clearly written and is quite succinct.

The authors state "we show for the first time, in a southern African setting, significant declines in the HIV incidence rate using data from a population-based cohort study," namely a 45% decline between 2012 and 2017. However, the incidence results confirm and add to findings from other population-based cohorts, and therefore are not the first. As the authors acknowledge, similar findings from population-based cohorts have been described for South Africa and Swaziland. The authors' claim would be more accurate if they described their method as prospective, eg a "prospectively followed population-based cohort" since the other studies in South Africa and Swaziland, as they

acknowledge, used cross-sectional methods. The authors may also want to discuss how their findings from Hlabisa fit into the context of the other studies and shed additional light. The list below illustrates how the current paper's findings confirm and add to the existing evidence that ART scale-up between 2011 and 2016 has led to remarkably similar declines in HIV incidence in several locations:

1. Rakai, Uganda: 2012-2016, 42% decline; open, prospective cohort, district-level; Grabowski et al., 2017
2. Western Kenya: 2011-2016, 50% decline; open, prospective cohort, district-level; Burgdorff et al., 2018
3. Swaziland: 2011-2016, 44% decline; cross-sectional, national cohort; Nkambule et al., IAS 2017
4. South Africa: 2012-2017, 44% decline, cross-sectional, national cohort; SABSSM, 2018
5. Hlabisa, KZN, Uganda: 2012-2017, 45% decline; open, prospective cohort, district-level; Vandormael et al. (current paper under review)

The authors do cite the South Africa and Swaziland surveys but have not selected the most suitable references. For South Africa, the authors cite "37. HSRC. HIV infections on decline, 2018" but a google search of these terms reveals "this page no longer exists". An alternative suggested citation is <http://www.hsrc.ac.za/en/media-briefs/hiv-aids-stis-and-tb/sabssm-launch-2018v2>

For Swaziland (now called Eswatini), the 2011 incidence estimate (Justman et al., 2017) is cited but the more suitable citation is the conference abstract that reports the incidence decline from 2011 to 2016: Nkambule et al., <http://programme.ias2017.org/Abstract/Abstract/5837>.

Despite not being the first, the paper will definitely be of interest to others in the field, given the longitudinal magnitude and rigor of the data and the analysis, and will influence thinking in the field primarily in that it confirms the remarkably similar findings by others, as described above.

There are some potential limitations that the authors do not discuss. For example, HIV testing is based on an algorithm that uses two ELISAs, a relatively old method that may have a higher rate of false positives than more current methods. Would be useful to know if the study team has used the exact same ELISA kits and algorithm throughout the duration of the study as this would confirm the rigor of the results. Strikingly absent from the paper are viral load data. If the study team has samples of blood in a repository that can be tested for viral load, that would add to the paper. The authors should add an explanation as to why viral load data are not described.

The manuscript would be improved with some additional minor revisions:

- o While the prevalence of participation, at 81%, is high, the authors should describe the characteristics of the 19% of eligible HIV-negative individuals who have not participated in the incidence testing/follow-up
- o Figure 3: color scheme for the various lines is complex and hard to follow and should be revised. For example, consider making HIV incidence the same color in both panels
- o Table 1: consider adding a footnote to explain why denominator for the year 2016 is so much lower than the denominator is in all the other years. Also incidence follow-up in 2016 was only 63%. Is this because data from 2016 are incomplete?
- o Table 2: suggest adding a footnote to the table to explain that the 2005 HIV incidence of 2.14 was used as the reference point estimate
- o One detail in the Methods section requires more detail. It's not clear if the authors have data about ART use that are specifically linked to the specific individuals who have participated in the household surveys (population-based ART data) or if the data describe only those individuals who are receiving health care services (facility-based ART data).

- o The analytic methods are generally sound. In describing the results, however, in particular age-adjusted HIV incidence, the text describes a "decline" in point estimates even when the 95% CIs overlap. For example: "Between 2012 and 2017, the HIV incidence rate (95% CI) among men declined by 61%, from 2.51 (1.87-3.38) to 0.97 (0.34-2.80) seroconversion events per 100 person-years." While this is a legitimate way to handle the results, for clarity and to avoid misleading less experienced readers, the authors should add text about which declines are significant and which ones are not.

Reviewer #1 (Remarks to the Author):

Given the relatively intractable nature of the HIV epidemic in SA, particularly in KZN, data showing positive effects of the ongoing roll out of combined prevention measures would represent an important and encouraging finding. Data presented here are from a large population based HIV surveillance cohort with over a decade of follow-up, and such cohorts represent the gold standard for assessing trends in HIV rates. However, there are a number of issues which complicate the interpretation of the findings, of which the principal ones are:

1. Given the changes in the rates of HIV testing over time, there is the potential for substantial selection bias. Information is needed on how the cohort's composition in each component (enumerated, eligible, contacted, tested, followed-up) has changed over time.

Response: We have undertaken several sensitivity analyses and provided additional information on the HIV testing rates to assess for the possibility of selection bias or confounding. Our findings were robust to these analyses. Specifically, we:

- 1) Included information on all participants that were eligible, contacted, and tested for HIV by year (Table 1).
- 2) Included information on the HIV testing rates by sex and age by year (Figure S1).
- 3) Included information on the mean age and sex of HIV-negative participants and repeat-testers (HIV incidence cohort) by year (Figure S2).
- 4) Included information on the average number of in- and out-migration events of all HIV testers by sex and year (Figure S2).
- 5) Included information on the HIV testing rate and mean age of men who reported being circumcised or not by year (Figure S3).
- 6) Included unadjusted and adjusted IRRs with the covariates age, marital status, condom use, circumcision status, household socio-economic status, migration history, and community HIV prevalence (Tables 2–3, Tables S1, S4–S6).
- 7) Included inverse probability weights in the Poisson regression models to control for potential selection biases associated with participant selection and drop-out (Tables S4–S6).
- 8) Included a sensitivity analysis which excludes repeat-testers (HIV cohort) with two or more consecutive missed test dates (Table S7).
- 9) Described how the single random-point imputation method, when coupled with a multiple imputation approach, is robust to the problem of consecutive missed tests (paragraph 7 of the *Discussion* section).

10) Added paragraphs 6 and 7 to the *Discussion* section to document these additions/sensitivity analyses.

In summary, our analyses show there was little change in the demographic composition (age and sex) of all HIV testers and repeat-testers over time. There was also little deviation in the in- and out-migration rates of participants that tested for HIV. Men who reported being circumcised had similar HIV testing rates and mean age to men who reported not being circumcised. There was little difference in the unadjusted IRRs (Tables 2-3, S1) and the adjusted IRRs (Tables S4–6) of the Poisson regression models. Similarly, there was little difference in the adjusted IRRs between all repeat-testers and the subset of repeat-testers with less than two consecutive missed test dates. Given these analyses, we find little evidence to suggest that the observed declines in HIV incidence can be explained by confounding or selection bias.

2. For all three models in Table 2, the declines in the adjusted incidence rate ratios are substantially more pronounced than one would expect from the actual incidence rates. For example, in Model 1, observed incidence declined from 3.48 in 2005-2007 to 3.35 in 2013-2017 (a very small change) but the adjIRR dropped sharply to 0.64. This suggests there could be substantial confounding. The authors should also present the unadjusted IRRs and include more information on how their adjustments were carried out and on the most likely sources of potential confounding.

Response: We apologize for the confusion. Following the Reviewer's recommendation, we present the unadjusted incidence rates (IRs) and unadjusted incidence rate ratios (IRRs) for men (Table 2) and women (Table 3). We slightly modified the ART coverage categories (Model 1) to more clearly demonstrate the declines in the HIV incidence. These results show a strong correspondence between the unadjusted IRs and unadjusted IRRs. We therefore find no suggestion of substantial confounding. We have included more information on how the adjustments were made and sources of potential confounding in paragraph 5 of the *Statistical Analysis* section.

3. Authorship list: It is surprising that the study team did not have an African collaborator (non-foreign, non-expatriate) who was sufficiently involved in study design, data collection, analyses and/or interpretation to be included among the authorship.

Response: The first and last authors are non-foreign, non-expatriate South Africans. Two of the middle authors are also South African permanent residents. The authors were heavily involved in many aspects of the study design, data collection and analysis, and interpretation of the study findings.

4. *Abstract: The referent year used in the tables and supplementary materials is 2006, not 2012, and I suggest that the incidence observed in that year also be used in the abstract (rather than 2012).*

Response: We appreciate this point but prefer to report the decline in the HIV incidence from its peak in 2012 to 2017 (a 43% reduction), which is the key finding of our study. This result corresponds with other population-based studies from sub-Saharan Africa that report similar declines in HIV incidence over the same period (see paragraph 4 of the *Discussion* section). We argue that the observed HIV incidence decline is consistent with the scale-up of prevention services in our study area, which began with the introduction of a voluntary medical male circumcision program in 2009 and changes to national ART eligibility criteria in 2010. We discuss this point in the first paragraph of the *Discussion* section.

5. *Page 3: "... the lack of population-based cohort studies..." and "previous estimates ...have been derived from mathematical models or cross-sectional assay-based studies" suggests there are no other such cohorts in Africa. Later on, the authors themselves cite a number of such studies. The term "... only a few other population-based cohort studies...." would be more appropriate.*

Response: We agree with the Reviewer and have revised the sentence to:

One major challenge in reliably measuring HIV incidence trends in southern Africa (as well as the broader African region) has been the limited number of population-based cohort studies.

6. *Page 4: HIV survey methods. The referenced paper (Ref 16, Int J Epi 2008) describing the survey reports information up to 2006/7 (12 years ago). Have the survey and outreach methods changed since then? For example, has HIV testing outreach been intensified, such that the characteristics of those being tested have changed? Reference 16 reports that the consent rate for HIV testing in 2005 and 2006 was only ~40%; the authors of this 2008 paper raise concerns about selection bias, and indicate that changes were being made both operationally and analytically to address this issue. This further underlines the need for detailed information on the actual composition of the cohort, by key characteristics, for each survey year, providing the reader with a sense of potential selection bias over time.*

Response: The survey methods did not change during the observation period: all households were visited each year by fieldworkers and the same HIV testing algorithm was used. For HIV incidence rate estimation, it is important that eligible HIV-negative participants have a first HIV-negative test followed by at least one valid HIV test result. We report that 76% of eligible HIV-negative participants had a repeat test and entered the HIV incidence cohort. In response to the Reviewer's comment, we write in paragraph 7 of the *Discussion* section:

We assessed if missed test dates could possibly have biased our HIV incidence rate results. To estimate the incidence rate, we identified uninfected participants with a first HIV-negative test result followed by at least one valid test result. We therefore considered 1) the percentage of eligible HIV-negative participants that entered into the HIV cohort and 2) the average length of the censoring interval, defined as the time between the latest HIV-negative and earliest HIV-positive test dates. First, we show that on average 76% of HIV-negative participants had a repeat test, which is relatively high inclusion rate for a prospective population-based cohort study. Second, we restricted our analysis to repeat-testers who missed no more than two consecutive test dates within the censoring interval. The results for the restricted and full cohorts are similar, as shown in Table S7. This is because the single random-point method, when coupled with standard multiple imputation procedures, produces robust incidence rate estimates—even with annual testing rates as low as 40% or average censored intervals as wide as four years. We have reported the theoretical and empirical basis for this result in recent work. Importantly, we show that the demographic composition (age and sex) of the HIV-negative testers and the HIV cohort remained stable throughout the observation period (see Figure S2). These sensitivity analyses, together with the single random-point approach for incidence rate estimation, constitute strong support for the robustness of our study findings.

And provide the reference to our work:

Vandormael A, Dobra A, Bärnighausen T, de Oliveira T, Tanser F. Incidence rate estimation, periodic testing and the limitations of the mid-point imputation approach. *International Journal of Epidemiology (IJE)*. 2017; 47(1):236-45.

To further assess possible selection biases, we:

- 1) Included information on all participants that were eligible, contacted, and tested for HIV by year (Table 1).
- 2) Included information on the HIV testing rates by sex and age by year (Figure S1).
- 3) Included information on the mean age and sex of HIV-negative participants and repeat-testers (HIV incidence cohort), and the average number of male and female in- and out-migration events of all HIV testers by year (Figure S2).
- 4) Included information on the HIV testing rate and mean age of men who reported being circumcised or not by year (Figure S3).
- 5) Included unadjusted and adjusted IRRs with the covariates age, marital status, condom use, circumcision status, household socio-economic status, migration history, and community HIV prevalence (Table S4–S6).
- 6) Included inverse probability weights in the Poisson regression models to control for potential selection biases associated with participant selection and drop-out (Tables S4–S6).

Given these analyses, we find little evidence to suggest that the observed declines in HIV incidence can be explained by confounding or selection bias related to missed tests.

7. Has any validation been carried out on the quality of the linkage of public health clinic data to AHRI data (missing data), and on the accuracy of the clinic data (the quality of clinical data collection and recording)?

Response: All data for this analysis comes from the AHRI demographic surveillance system. The only clinic data we used were the ART initiation and clinic visit dates, as shown in Model 1 of Tables 2–3. As far as we know, no formal analysis has been undertaken on the quality of linkage to the clinic data. However, it is unlikely that missing ART data will have dramatically affected our results. Nevertheless, we have discussed this as a possible limitation in paragraph 8 of the *Discussion* section.

8. Why is the age range for men and women different?

Response: When the HIV surveillance system started in 2004, the burden of HIV was highest among younger women and highest among older men. For this reason, the male age range was extended to 54 years. In the second sentence of the *Statistical Analysis* section, we now write “These age ranges were used to ensure consistency with previous AHRI analyses” and provide the relevant citations in the revised manuscript.

9. Page 5: Are participants included if they have missed one or more surveys, and if so, how many such skips are allowed? What proportion of persons have no, one, two or more missing time points? Imputation would suggest that the timing of incidence is random within a given time frame, but this assumption might not hold true over long periods of time, if there are significant secular changes (for example hypothetically, a large in-migration event which brings in a substantial group of high risk individuals in a given year). Supplementary data on observed HIV incidence rates allowing for only one or two missed intervals would be useful.

Response: We included all repeat-testers in the HIV cohort if they had a first HIV-negative test result followed by at least one subsequent HIV test result. The issue of missed test dates is a major area of our research, see:

Vandormael A, et al. Incidence rate estimation, periodic testing and the limitations of the mid-point imputation approach. *IJE*, 2017; 47(1):236-45.

which we cite extensively in response to the Reviewer’s comment here and elsewhere. In short, consecutive missed tests are a problem if one imputes the unobserved infection date at the mid-point of the latest HIV-negative and earliest HIV-positive test dates. (Mid-point imputation is a popular *ad hoc* imputation approach which we do not recommend.) Consecutive missed test

dates are not a severe problem if one uses the single random-point method and standard multiple imputation procedures. It is for this reason that we generated 300 imputed datasets and used Rubin's rules to obtain estimates and standard errors. (Also note that an HIV-negative participant remains negative, no matter how many missed [consecutive or otherwise] test dates between the earliest HIV-negative and latest HIV-negative test dates.)

In our IJE paper, we demonstrate the accuracy and robustness of the single random-point method both theoretically and empirically. In our simulation studies, we show that the single random-point method produces incidence rate estimates close to the true incidence rate, even if the annual HIV testing rate is as low as 40% or the average length of the censored interval is as wide as 4 years. Nevertheless, following the Reviewer's suggestion, we undertook a sensitivity analysis by restricting the data to only repeat-testers with no more than two consecutive missed test dates during the censoring interval. We summarized these points in paragraph 7 of the *Discussion* section:

We assessed if missed test dates could possibly have biased our HIV incidence rate results. To estimate the incidence rate, we identified uninfected participants with a first HIV-negative test result followed by at least one valid test result. We therefore considered 1) the percentage of eligible HIV-negative participants that entered into the HIV cohort and 2) the average length of the censoring interval, defined as the time between the latest HIV-negative and earliest HIV-positive test dates. First, we show that on average 76% of HIV-negative participants had a repeat test, which is relatively high inclusion rate for a prospective population-based cohort study. Second, we restricted our analysis to repeat-testers who missed no more than two consecutive test dates within the censoring interval. The results for the restricted and full cohorts are similar, as shown in Table S7. This is because the single random-point method, when coupled with standard multiple imputation procedures, produces robust incidence rate estimates—even with annual testing rates as low as 40% or average censored intervals as wide as four years. We have reported the theoretical and empirical basis for this result in recent work. Importantly, we show that the demographic composition (age and sex) of the HIV-negative testers and the HIV cohort remained stable throughout the observation period (see Figure S2). These sensitivity analyses, together with the single random-point approach for incidence rate estimation, constitute strong support for the robustness of our study findings.

We therefore keep the full cohort of repeat-testers (irrespective of missed tests) for our final analyses. Finally, Figure S2 shows there were no large in-migration events (relative to out-migration events or relative to other years) during the observation period.

10. Does the interview collect information on numbers and types of partners since the last survey?

Response: Yes, this data is collected by field workers and has been used in previously analyses:

Tanser, F., et al. 2012. *Effect of concurrent sexual partnerships on rate of new HIV infections in a high-prevalence, rural South African population: a cohort study*. The Lancet 378 (9787): 247-255.

11. Page 6 (Results): What were the criteria that rendered 23% of those enumerated ineligible for HIV testing? The proportion of persons contacted and consenting to be tested increased from 38% to 75% over time: key characteristics of persons actually tested need to be provided for each time period.

Response: We now include a definition for eligibility criteria in the *Statistical Analysis* section and footnote of Table 1, where we write:

Eligibility was defined as being >15 years of age, mentally able, and a household resident in the last 12 months.

The cumulative increase in participants having tested at least once for HIV is due to the fact that the probability of being captured for at least one HIV test increases with each additional survey round/year. We clarify this point in footnote 1 of the Table 1 in the revised manuscript.

Following the Reviewer's recommendation, we also:

- 1) Included information on all participants that were eligible, contacted, and tested for HIV by year (Table 1).
- 2) Included information on the HIV testing rates by sex and age by year (Figure S1).
- 3) Included information on the mean age and sex of HIV-negative participants and repeat-testers (HIV incidence cohort), and the average number of male and female in- and out-migration events of all HIV testers by year (Figure S2).
- 4) Included information on the HIV testing rate and mean age of men who reported being circumcised or not by year (Figure S3).
- 5) Included unadjusted and adjusted IRRs with the covariates age, marital status, condom use, circumcision status, household socio-economic status, migration history, and community HIV prevalence (Tables S4-S6).
- 6) Included inverse probability weights in the Poisson regression models to control for potential selection biases associated with participant selection and drop-out (Tables S4-S6).
- 7) Included a sensitivity analysis which excludes repeat-testers (HIV cohort) with two or more consecutive missed test dates (Table S7).
- 8) Described how the single random-point imputation method, coupled with a multiple imputation approach, is robust to the problem of consecutive missed tests (paragraph 7 of the *Discussion* section).
- 9) Added two full paragraphs to the *Discussion* section to document these additional sensitivity analyses.

Given these analyses, we find little evidence to suggest that the observed declines in HIV incidence can be explained by confounding or selection bias.

12.1. Page 7 and Figure 4: The text and figure contradict each other. The text reports that HIV incidence among circumcised men fell from 2.2 to 0.03 per 100 py: This is probably a typo, since the figure shows the decline to achieve ~0.3/100 py. However, the footnote in the figure says HIV incidence among circumcised men fell by 99%, which based on reports by others seems very optimistic (the actual figure suggests the reduction was in the order of ~85%).

Response: We apologize for the confusion and have fixed this issue in both the main text and the footnote to Figure 4.

12.2 It is puzzling that a steady decline in incidence among circumcised men began in 2009 whereas in the uncircumcised, incidence actually went up slightly between 2009 and 2013. A circumcised male experiences a fairly constant ~50-60% reduction in risk of HIV. Presumably, the additional decline in incidence over time in circumcised men occurs because of the protective effects of increasing rates of female partner ART use. One would expect the uncircumcised men to also show some declines in incidence between 2009-2013, as more women went on ART.

Response: Yes, but HIV incidence among uncircumcised men only declined once female ART coverage surpassed 35% in 2012. Before 2012, the slight increase in incidence among uncircumcised men is consistent with the slight increase in the overall HIV incidence rate during this time (see for example Figure 1). Our results therefore suggest that the scale-up of VMMC in 2009 had an earlier impact on the HIV incidence rate of circumcised men when compared with uncircumcised men. We now write in the third paragraph of the *Discussion* section that:

The lagged decline in HIV incidence among uncircumcised men when compared with circumcised men, coincides with an increase in CD4+ T-cell count eligibility criteria from <200 to <350 cells/ μ L in 2011 and female ART coverage surpassing 35% in 2012.

In the footnote to Figure 4, we also write:

Shows the HIV incidence rate among circumcised men declined steadily between 2009 and 2017, from 1.67 to 0.42 events per 100 person-years—a 75% decline. Among men reporting being uncircumcised, the age-adjusted HIV incidence increased slightly before 2012, which is consistent with the overall increase in HIV incidence during this period. Following the introduction of more inclusive ART eligibility criteria and female ART coverage surpassing 35%, the age-adjusted HIV incidence among uncircumcised men declined from 2.78 to 1.64 events per 100 person-years between 2012 and 2017—a 41% decline.

12.3 It would be good to present data on circumcised and uncircumcised male characteristics over time, in order to explore if selection bias may have affected the results.

Response: We agree and now present the HIV testing rates and mean age of men reporting being circumcised or not in Figure S3. The analysis shows little difference between these two groups. It is unlikely that the differences in HIV incidence between circumcised and uncircumcised men is being driven by selection biases.

13. Page 8: Since ART first became available in 2004, and expanded over time, the comparison should be between that 2004 and 2017. Selecting the year with the highest observed incidence rate (2012) may overstate the effects of combined HIV prevention in this population.

Response: We respectfully disagree with the Reviewer. The intent of our article is to assess the impact of HIV prevention services, which were largely started after 2009. These include more relaxed ART eligibility criteria, expanded test and treat services, and the introduction of a male medical circumcision program. While comparing HIV incidence between 2005 and 2017 would assess for changes over the entire duration of the population cohort, it would not accurately measure the effect of prevention services which began more recently. Further, selecting this period makes our findings—a 43% reduction in the HIV incidence rate—comparable with other population-based studies from sub-Saharan Africa over the same period. We discuss this point in the fourth paragraph of the *Discussion* section.

14. The supplementary materials should include a detailed table indicated key characteristics of the enumerated, contacted and tested population for each study year, as well as characteristics of those followed up.

Response: Following the Reviewer's suggestions, we have:

- 1) Included information on all participants that were eligible, contacted, and tested for HIV by year (Table 1).
- 2) Included information on the HIV-negative participants that were eligible for entry into the HIV cohort and the number/percentage of these HIV-negative participants that had a repeat-test (and therefore entered into the HIV cohort) (Table 1).
- 3) Included information on the HIV testing rates by sex and age by year (Figure S1).
- 4) Included information on the mean age and sex of HIV-negative participants and repeat-testers (HIV incidence cohort), and the average number of male and female in- and out-migration events for all HIV testers by year (Figure S2).
- 5) Included information on the HIV testing rate and mean age of men who reported being circumcised or not by year (Figure S3).

In summary, our analyses show that the HIV testing rate among participants by sex and age remained relatively stable over time. The demographic composition of HIV testers and repeat-

testers did not change over the observation period, with little deviation in the in- and out-migration rates of all HIV testers. HIV testing and age characteristics were similar among men reporting circumcision or not. Given these analyses, we find little evidence to suggest that the observed declines in HIV incidence can be explained by confounding or selection bias.

15. *Table 1. Please define what is meant by “eligible for testing” (who is not eligible?)*

Response: We apologize for the confusion and define eligibility in the *Statistical Analysis* section and the footnote of Table 1, where we write:

Eligibility was defined as being >15 years of age, mentally able, and a household resident in the last 12 months.

16. *If I understand correctly, the “ever tested” rate includes anyone who has ever had an HIV test since 2005. It would be informative to know what proportion of previously HIV-neg participants received HIV re-testing in each year. How did the characteristics of those tested in each year differ over time?*

Response: We agree with the Reviewer. In Table 1, we now present this information in the columns “HIV-negative Tested” (the number of HIV-negative participants that were eligible for entry into the HIV cohort) and Repeat-testers” (the number and percentage of these participants that repeat-tested and entered into the HIV cohort). We also include in Figure S2 information by year on the mean age and proportion of HIV-negative testers and repeat-testers. The figure shows that these demographic characteristics did not markedly differ over the observation period. We write in paragraph 7 of the *Discussion* section:

Importantly, we also show that the demographic composition (age and sex) of the HIV-negative testers and the HIV cohort remained stable throughout the observation period (Figure S2). These sensitivity analyses, together with the single random-point approach for incidence rate estimation, constitute strong support for the robustness of our study findings.

17. *Are “ever tested” rates based on self-report, AHRI survey data, or from clinic data? If it is the latter, has the quality of the clinic linkage and clinic data been verified?*

Response: The ever-tested rates are based on AHRI HIV survey data. We now clarify this point in the revised footnote of Table 1, where we write:

Since 2005, trained field workers have visited all households in the AHRI surveillance area to undertake an annual HIV survey. After obtaining written consent, the field workers take dried blood spot samples for HIV testing. Field workers also collect data on whether participants were contact and tested.

18. *The legend refers to “contacted from eligible” but the footnote defines this as “consented to*

an HIV test”. If it is the latter, why are the proportions “ever tested” so much lower than those consented?

Response: We apologize for the confusion and have revised Table 1 to improve clarity. The revised footnote in Table 1 now correctly indicates “contacted from eligible”. The ever-tested percentage is lower than the contacted percentage because some proportion of contacted participants refuse to test, which we show in Table 1.

19. Are participants allowed one or more “skipped” follow-up survey visits in the calculation of incidence rates?

Response: Yes, we included all repeat-testers in the HIV cohort if they had a first HIV-negative test result followed by at least one subsequent HIV test result. Following the Reviewer’s suggestion, we undertook a sensitivity analysis by excluding any repeat-tester with two or more consecutive missed test dates between the latest HIV-negative and earliest HIV-positive test dates. We now write in paragraph 7 of the *Discussion* section:

We assessed if missed test dates could possibly have biased our HIV incidence rate results. To estimate the incidence rate, we identified uninfected participants with a first HIV-negative test result followed by at least one valid test result. We therefore considered 1) the percentage of eligible HIV-negative participants that entered into the HIV cohort and 2) the average length of the censoring interval, defined as the time between the latest HIV-negative and earliest HIV-positive test dates. First, we show that on average 76% of HIV-negative participants had a repeat test, which is relatively high inclusion rate for a prospective population-based cohort study. Second, we restricted our analysis to repeat-testers who missed no more than two consecutive test dates within the censoring interval. The results for the restricted and full cohorts are similar, as shown in Table S7. This is because the single random-point method, when coupled with standard multiple imputation procedures, produces robust incidence rate estimates—even with annual testing rates as low as 40% or average censored intervals as wide as four years. We have reported the theoretical and empirical basis for this result in recent work. Importantly, we show that the demographic composition (age and sex) of the HIV-negative testers and the HIV cohort remained stable throughout the observation period (see Figure S2). These sensitivity analyses, together with the single random-point approach for incidence rate estimation, constitute strong support for the robustness of our study findings.

20. Why is the 2016 population of individuals enumerated, eligible and contacted substantially lower than in prior years? I understand that the 2017/2018 data are incomplete, but 2016 should be “in the can”.

Response: Following the latest data release, this 2016 issue has been corrected in Table 1, which now includes information from 2017.

21. *A flow diagram could be very useful to clearly show who drops out where in the surveillance “cascade” (from enumerated to receipt of an HIV re-test) and how that has changed over time.*

Response: Such an analysis has been undertaken in great detail by:

Larmarange J, Mossong J, Bärnighausen T, Newell ML. Participation dynamics in population-based longitudinal HIV surveillance in rural South Africa. *PloS One*. 2015; 10(4):e0123345.

which we cite in the revised manuscript. Following the Reviewer’s suggestion, we have also:

- 1) Included information on all participants that were eligible, contacted, and tested for HIV by year (Table 1).
- 2) Included information on the HIV testing rates by sex and age by year (Figure S1).
- 3) Included information on the mean age and sex of HIV-negative participants and repeat-testers (HIV incidence cohort), and the average number of male and female in- and out-migration events by year (Figure S2).
- 4) Included information on the HIV testing rate and mean age of men who reported being circumcised and not circumcised by year (Figure S3).
- 5) Included unadjusted and adjusted IRRs with the covariates age, marital status, condom use, circumcision status, household socio-economic status, migration history, and community HIV prevalence (Tables S4-S6).
- 6) Included a sensitivity analysis which excludes repeat-testers (HIV cohort) with two or more consecutive missed test dates (Table S7).
- 7) Added two paragraphs to the *Discussion* section to document these additional sensitivity analyses.

22. *Table 2: incidence rate trends and IRRs by ART coverage. As I commented in the introduction, for all three models in Table 2, the declines in the adjusted incidence rate ratios are substantially more pronounced than one would expect from the actual incidence rates. This suggests there could be substantial confounding. The authors should also present the unadjusted IRRs and include more information on potential confounders.*

Response: We apologize for the confusion. Following the Reviewer’s recommendation, we present the unadjusted incidence rates (IRs) and unadjusted incidence rate ratios (IRRs) for men (see Table 2) and women (see Table 3). We also slightly modified the ART coverage categories (Model 1) to more clearly demonstrate the declines in the HIV incidence. Overall, there is little difference between the unadjusted IRs and unadjusted IRRs. Based on these results, we find no suggestion of substantial confounding. We have included more information on how the

adjustments were made and sources of potential confounding in paragraph 5 of the *Statistical Analysis* section.

23. In addition, the ART coverage threshold by 2010-2012 was already 20-39%, compared to prior time intervals with much lower coverage, yet the observed incidence rate in 2010-2012 was higher than in any other period (even as one would expect the rate to start edging down, given the expanded ART coverage). What might account for this: secular factors, differences in the characteristics of those who provided samples for HIV testing, other?

Response: We thank the Reviewer for this comment. In paragraph 6 of the Discussion section, we now write:

We note that the unadjusted IRRs declined once ART coverage surpassed 35%, suggesting that treatment uptake had to reach a population threshold before reductions in HIV incidence could be observed. After adjusting for HIV prevalence and other well-established risk factors, we found that increased ART coverage was associated with monotonic declines in the HIV acquisition risk. In earlier analyses, we demonstrated that reductions in the HIV acquisition risk were associated with increased ART coverage at the community, household, and serodiscordant couple levels, holding HIV prevalence and other well-established risk factors constant. Given the correspondence between the unadjusted and adjusted IRRs, and between our results and previous research, it is unlikely that the observed declines in HIV incidence can be explained by model selection or confounding.

24. S1: It would be more intuitive to show uncircumcised men as the referent group: this would be consistent with how the VMMC randomized trials presented their data (i.e., would more readily indicate the degree to which VMMC is protective in the KZN population cohort).

Response: Following the Reviewer's recommendation, we now use uncircumcised men as the referent group in Table S4.

25. S1-S3: Please specify whether the reported rate ratios are adjusted or unadjusted.

Response: This has been done.

26. S1 and S2: Why do the numbers of person-years progressively decline over time? In 2017, the reported persons years represent only 45% of those in 2016: this implies a lower follow-up rate compared to the 63.4% reported in Table 1. To reiterate a prior question, are participants allowed skips in follow-up and if so, how many? Is there a difference in HIV incidence between those followed in consecutive surveys compared to those who skip rounds?

Response: This issue has been addressed in Table 1. We included all repeat-testers in the HIV cohort if they had a first HIV-negative test result followed by at least one subsequent HIV test

result. Following the Reviewer's suggestion, we undertook a sensitivity analysis by excluding any repeat-tester with two or more consecutive missed test dates. We now write in paragraph 7 of the *Discussion*:

We assessed if missed test dates could possibly have biased our HIV incidence rate results. To estimate the incidence rate, we identified uninfected participants with a first HIV-negative test result followed by at least one valid test result. We therefore considered 1) the percentage of eligible HIV-negative participants that entered into the HIV cohort and 2) the average length of the censoring interval, defined as the time between the latest HIV-negative and earliest HIV-positive test dates. First, we show that on average 76% of HIV-negative participants had a repeat test, which is relatively high inclusion rate for a prospective population-based cohort study. Second, we restricted our analysis to repeat-testers who missed no more than two consecutive test dates within the censoring interval. The results for the restricted and full cohorts are similar, as shown in Table S7. This is because the single random-point method, when coupled with standard multiple imputation procedures, produces robust incidence rate estimates—even with annual testing rates as low as 40% or average censored intervals as wide as four years. We have reported the theoretical and empirical basis for this result in recent work. Importantly, we show that the demographic composition (age and sex) of the HIV-negative testers and the HIV cohort remained stable throughout the observation period (see Figure S2). These sensitivity analyses, together with the single random-point approach for incidence rate estimation, constitute strong support for the robustness of our study findings.

27. It is intriguing that the communities with the highest cumulative out-migration rate would also have the highest incidence. Are these highly mobile communities overall (did they also have the highest in-migration rates, potentially receiving more individuals who are not yet accessing prevention and treatment services?)

Response: Yes, in our surveillance area, those participants who migrate frequently and for extended periods of time are at highest risk of HIV acquisition. We cite the two main studies on migration frequency and cumulative out-migration in the revised manuscript.

McGrath N, Eaton JW, Newell ML, Hosegood V. Migration, sexual behaviour, and HIV risk: a general population cohort in rural South Africa. *The lancet HIV*. 2015 Jun 1;2(6):e252-9.

Dobra A, Bärnighausen T, Vandormael A, Tanser F. Space-time migration patterns and risk of HIV acquisition in rural South Africa. *AIDS (London, England)*. 2017 Jan 2;31(1):137.

28. S6: Although the legend refers to the ART coverage threshold, it would seem that only the

first section of the table is related to actual ART coverage, while the incidence rates by other characteristics are for all time intervals combined.

Response: Yes, this is correct. We only use the ART coverage categories for Model 1 of the Poisson regression analyses.

Reviewer #2 (Remarks to the Author):

The paper describes a 45% decline over time, between 2005-2015, in HIV incidence in the Hlabisa sub-district of KwaZulu-Natal region of South Africa, based on an open, longitudinal, population based cohort that has undergone approximately annual rounds of HIV testing. The methods and analytic approach are generally very rigorous, and manuscript is very clearly written and is quite succinct.

1. The authors state "we show for the first time, in a southern African setting, significant declines in the HIV incidence rate using data from a population-based cohort study," namely a 45% decline between 2012 and 2017. However, the incidence results confirm and add to findings from other population-based cohorts, and therefore are not the first. As the authors acknowledge, similar findings from population-based cohorts have been described for South Africa and Swaziland. The authors' claim would be more accurate if they described their method as prospective, eg a "prospectively followed population-based cohort" since the other studies in South Africa and Swaziland, as they acknowledge, used cross-sectional methods.

Response: We agree and have made this specific change in the *Introduction*, where we write:

One major challenge in reliably measuring HIV incidence trends in southern Africa (as well as the broader African region) has been the limited number of prospectively followed, population-based cohort studies.

2. The authors may also want to discuss how their findings from Hlabisa fit into the context of the other studies and shed additional light. The list below illustrates how the current paper's findings confirm and add to the existing evidence that ART scale-up between 2011 and 2016 has led to remarkably similar declines in HIV incidence in several locations:

- 1. Rakai, Uganda: 2012-2016, 42% decline; open, prospective cohort, district-level; Grabowski et al., 2017*
- 2. Western Kenya: 2011-2016, 50% decline; open, prospective cohort, district-level; Burgdorff et al., 2018*
- 3. Swaziland: 2011-2016, 44% decline; cross-sectional, national cohort; Nkambule et al., IAS 2017*
- 4. South Africa: 2012-2017, 44% decline, cross-sectional, national cohort; SABSSM, 2018*
- 5. Hlabisa, KZN, Uganda: 2012-2017, 45% decline; open, prospective cohort, district-level; Vandormael et al. (current paper under review)*

The authors do cite the South Africa and Swaziland surveys but have not selected the most

suitable references. For South Africa, the authors cite “37. HSRC. HIV infections on decline, 2018” but a google search of these terms reveals “this page no longer exists”. An alternative suggested citation is <http://www.hsrc.ac.za/en/media-briefs/hiv-aids-stis-and-tb/sabssm-launch-2018v2>

For Swaziland (now called Eswatini), the 2011 incidence estimate (Justman et al., 2017) is cited but the more suitable citation is the conference abstract that reports the incidence decline from 2011 to 2016: Nkambule et al., <http://programme.ias2017.org/Abstract/Abstract/5837>.

Response: We thank the Reviewer for these suggestions, and now include the HSRC and Nkambule et al. references in the revised manuscript.

3. Despite not being the first, the paper will definitely be of interest to others in the field, given the longitudinal magnitude and rigor of the data and the analysis, and will influence thinking in the field primarily in that it confirms the remarkably similar findings by others, as described above. There are some potential limitations that the authors do not discuss. For example, HIV testing is based on an algorithm that uses two ELISAs, a relatively old method that may have a higher rate of false positives than more current methods. Would be useful to know if the study team has used the exact same ELISA kits and algorithm throughout the duration of the study as this would confirm the rigor of the results.

Response: We agree. At the end of the *HIV Survey Methods* section, we now write:

The same HIV testing algorithm was used throughout the observation period.

4. Strikingly absent from the paper are viral load data. If the study team has samples of blood in a repository that can be tested for viral load, that would add to the paper. The authors should add an explanation as to why viral load data are not described.

Response: Following the Reviewer’s recommendation, we now write in the *HIV Survey Methods* section:

From all HIV-positive test results in 2011, 2013, and 2014, we obtained viral load measurements and used a threshold of 1,550 copies/mL to identify participants with detectable viremia. We then calculated population-based estimates of the prevalence of detectable viremia for these three survey years, as described in greater detail elsewhere.

And provide the relevant citation:

Vandormael A, Bärnighausen T, Herbeck J, Tomita A, Phillips A, Pillay D, de Oliveira T, Tanser F. Longitudinal trends in the prevalence of detectable HIV viremia: population-based evidence from rural KwaZulu-Natal, South Africa. *Clinical Infectious Diseases*. 2017; 66(8):1254-60.

Plans are underway to obtain viral load measurements for the 2015–2017 survey years. We also report the male and female prevalence of detectable viremia in the *Results* section and in Figure 3. In the second paragraph of the *Discussion* section, we write:

At least partly due to perinatal HIV screening and treatment programs, women are more likely than men to test for HIV, initiate ART early, and achieve long-term undetectable viremia. Given the generalized, heterosexual HIV epidemic in sub-Saharan Africa, this means that men will have a comparatively lower risk of meeting female sexual partners with detectable viremia and acquiring HIV (on average and under the assumption of a well-mixed population sexual network). We have previously found that the prevalence of detectable viremia, when combined with the underlying spatial variation in the proportion of the population infected (HIV prevalence), was strongly associated with the future HIV infections. Specifically, every 1% increase in the overall proportion of a population having detectable viremia was independently associated with a 6.3% increase in an individual's risk of HIV acquisition. The earlier declines in male HIV incidence are therefore broadly consistent with higher female ART coverage and lower female prevalence of detectable viremia in the AHRI surveillance area.

and provide the citation to:

Tanser F, Vandormael A, Cuadros D, Phillips AN, de Oliveira T, Tomita A, Bärnighausen T, Pillay D. Effect of population viral load on prospective HIV incidence in a hyperendemic rural African community. *Science translational medicine*. 2017; 9(420):eaam8012.

5. The manuscript would be improved with some additional minor revisions: While the prevalence of participation, at 81%, is high, the authors should describe the characteristics of the 19% of eligible HIV-negative individuals who have not participated in the incidence testing/follow-up.

Response: We have undertaken several sensitivity analyses and provided additional information on the HIV testing rates to assess for the possibility of selection bias or confounding. Our findings were robust to these analyses. Specifically, we:

- 1) Included information on all participants that were eligible, contacted, and tested for HIV by year (Table 1).
- 2) Included information on the HIV testing rates by sex and age by year (Figure S1).
- 3) Included information on the mean age and sex of HIV-negative participants and repeat-testers (HIV incidence cohort) by year (Figure S2).
- 4) Included information on the average number of in- and out-migration events of all HIV testers by sex and year (Figure S2).
- 5) Included information on the HIV testing rate and mean age of men who reported being circumcised or not by year (Figure S3).

- 6) Included unadjusted and adjusted IRRs with the covariates age, marital status, condom use, circumcision status, household socio-economic status, migration history, and community HIV prevalence (Tables 2–3, Tables S1, S4–S6).
- 7) Included inverse probability weights in the Poisson regression models to control for potential selection biases associated with participant selection and drop-out (Tables S4–S6).
- 8) Included a sensitivity analysis which excludes repeat-testers (HIV cohort) with two or more consecutive missed test dates (Table S7).
- 9) Described how the single random-point imputation method, when coupled with a multiple imputation approach, is robust to the problem of consecutive missed tests (paragraph 7 of the *Discussion* section).
- 10) Added paragraphs 6 and 7 to the *Discussion* section to document these additions/sensitivity analyses.

In summary, our analyses show there was little change in the demographic composition (age and sex) of all HIV testers and repeat-testers over time. There was also little deviation in the in- and out-migration rates of participants that tested for HIV. Men who reported being circumcised had similar HIV testing rates and mean age to men who reported not being circumcised. There was little difference in the unadjusted IRRs (Tables 2-3, S1) and the adjusted IRRs (Tables S4–6) of the Poisson regression models. We also write in paragraph 7 of the *Discussion* section:

We assessed if missed test dates could possibly have biased our HIV incidence rate results. To estimate the incidence rate, we identified uninfected participants with a first HIV-negative test result followed by at least one valid test result. We therefore considered 1) the percentage of eligible HIV-negative participants that entered into the HIV cohort and 2) the average length of the censoring interval, defined as the time between the latest HIV-negative and earliest HIV-positive test dates. First, we show that on average 76% of HIV-negative participants had a repeat test, which is relatively high inclusion rate for a prospective population-based cohort study. Second, we restricted our analysis to repeat-testers who missed no more than two consecutive test dates within the censoring interval. The results for the restricted and full cohorts are similar, as shown in Table S7. This is because the single random-point method, when coupled with standard multiple imputation procedures, produces robust incidence rate estimates—even with annual testing rates as low as 40% or average censored intervals as wide as four years. We have reported the theoretical and empirical basis for this result in recent work. Importantly, we show that the demographic composition (age and sex) of the HIV-negative testers and the HIV cohort remained stable throughout the observation period (see Figure S2). These sensitivity analyses, together with the single random-point approach for incidence rate estimation, constitute strong support for the robustness of our study findings.

Given these analyses, we find little evidence to suggest that the observed declines in HIV incidence can be explained by confounding or selection bias.

6. *Figure 3: color scheme for the various lines is complex and hard to follow and should be revised. For example, consider making HIV incidence the same color in both panels.*

Response: We agree and have made this change to Figure 3.

7. *Table 1: consider adding a footnote to explain why denominator for the year 2016 is so much lower than the denominator is in all the other years. Also incidence follow-up in 2016 was only 63%. Is this because data from 2016 are incomplete?*

Response: We agree, this issue and has been fixed in the latest data release.

8. *Table 2: suggest adding a footnote to the table to explain that the 2005 HIV incidence of 2.14 was used as the reference point estimate*

Response: We agree and have revised the Table 2–3 footnotes (and elsewhere) to better explain the reference categories.

9. *One detail in the Methods section requires more detail. It's not clear if the authors have data about ART use that are specifically linked to the specific individuals who have participated in the household surveys (population-based ART data) or if the data describe only those individuals who are receiving health care services (facility-based ART data).*

Response: This is technically population-based ART data. Following the Reviewer's comment, we now write in the *Statistical Analysis* section:

Approximately 60% of all HIV-positive participants in the demographic surveillance system have been linked to the public health care clinics, which enabled us to obtain information on ART initiation and clinic visit dates. We assumed the remaining participants were ART-naïve or initiated ART at other public health-care clinics.

10. *The analytic methods are generally sound. In describing the results, however, in particular age-adjusted HIV incidence, the text describes a "decline" in point estimates even when the 95% CIs overlap. For example: "Between 2012 and 2017, the HIV incidence rate (95% CI) among men declined by 61%, from 2.51 (1.87-3.38) to 0.97 (0.34-2.80) seroconversion events per 100 person-years." While this is a legitimate way to handle the results, for clarity and to avoid misleading less experienced readers, the authors should add text about which declines are significant and which ones are not.*

Response: This problem with the 95% CI overlap has been addressed following the latest data release. P-values for the unadjusted and adjusted Incidence Rate Ratios (IRR) are presented in Table 2–3 and Tables S1, S4–S6.

Reviewers' comments:

Reviewer #1 (Remarks to the Author):

The authors have been very responsive in providing additional data and explanations, bolstering their case that significant declines in HIV incidence are occurring in their study population.

I have only a few more comments:

1. Why was the detectable viral load set at >1,550 copies/mL, rather than the more standard >1,000 copies used by the WHO? What are the rates of suppression using the WHO definition?

2. The unadjusted IRRs and the adjusted IRRs (for example, Tables 2 and S4 for men) should ideally be shown on the same table, to enable direct comparison between unadjusted and adjusted results. Also, the adjusted information is important, and deserves greater prominence than it might receive if it is solely in the supplementary materials.

Reviewer #2 (Remarks to the Author):

Comments on revised manuscript, NCOMMS-19-13204A, Vandormael et al

The authors have done a thorough job in responding to earlier comments. A few additional comments on the revised version are below.

Main comments:

Results:

-Para 1, line 4: would insert text to make it explicit that an average of 35% of all eligible enumerated adult participants in Hlabisa took part in HIV testing each year (0.85×0.41).

-Para 2, line 1 (and Table 1): Origin of the study group of 22,239 participants is not clear. Authors refer to another paper Larmarange in the response document (response #21) and comment that the flow/consort diagram that would explain the study group is now cited in the revised manuscript but in fact this citation is not in the Reference list. Since this study group is central to this paper, authors should briefly explain it here.

-On a related note, origin of all of the values in Table 1, column 11 are not clear; it is puzzling that all of the values in column 11 are consistently greater than the numerators listed in column 9. This needs clarification.

-Para 8 on MV regression: the declines do not look monotonic to me. They are larger and more consistent for the men than for the women so they might be described as monotonic among the men but I would not use that term for the declines seen among the women. Also Model 1 has only 3 values so hard to call any pattern monotonic. Same point re "monotonic" for para 6 of Discussion

Disc:

-Suggest authors provide context on whether and how well data from one-third of Hlabisa adult population represent all of Hlabisa. And how well do data from Hlabisa represent all of SA or other regions of southern Africa, especially given rural predominance.

Tables and Figures:

-Table 1, as noted above: Column 9 numerator values are consistently smaller than column 11 values—please explain in Methods and or with a table footnote.

-Table 1: Would also add Average of Total row at bottom of the table since the text in the Results section refers to these average values.

-Table S4 and S5: add reference group for age bands (ie ages 15-19 y). Specify if HIV prevalence refers to HIV prevalence among opposite sex

Table S6: Does HIV prevalence here refer to total in Hlabisa community?

Minor comments:

Methods, Statistical analysis:

-First two paragraphs are more about survey methods than about statistical methods.

-Why use 1550 c/mL as a cut point? This is not the usual convention and while the authors refer to their prior work (ref 22) as an explanation, a brief sentence to justify the cut point would be helpful to the reader.

Results:

-Para 5: Text on detectable viremia should refer to Figure 3 to illustrate the data points described. Just need to add it in parentheses at right location.

-Para 9 on MV regression: "...and women had a significantly higher adjusted IRR..." This refers to Table S6 which does not list Women as a variable; is this sentence referring to the "not married" variable?

-Figure 1: Would say that incidence began to decline "after" 2012 (and after 2014 for women) rather than "from 2012" or "from 2014", as the latter suggests the decline became evident starting in 2012. This is minor semantic point but it took me some time to see the authors mean the decline is evident after 2012/2014.

-Fig 4: would add 95% CIs, as in Figure 1

20 September 2019

We thank both reviewers for their effort and time in reviewing this manuscript. Their comments have led to significant improvements in the manuscript's clarity and quality. We address each of the comment's below. Changes to the main text are highlighted in red.

Reviewer #1 (Remarks to the Author):

The authors have been very responsive in providing additional data and explanations, bolstering their case that significant declines in HIV incidence are occurring in their study population.

I have only a few more comments:

1. Why was the detectable viral load set at >1,550 copies/mL, rather than the more standard >1,000 copies used by the WHO? What are the rates of suppression using the WHO definition?

Response: The viral load level of 1,550 copies/mL is the lowest detection limit of the quantification method used by the AHRI laboratory for dried blood spots, which was standard for the field at the time of testing. Unfortunately, for this reason, we cannot quantify suppression rates using a lower (WHO) detection limit of 1,000 copies/mL. In the revised manuscript we now write in paragraph 3 of the *Demographic and HIV survey* section:

Viral load levels have been obtained from all HIV-positive samples collected from the HIV survey in 2011, 2013, and 2014. Nucleic acid was extracted with NucliSENS EasyMag (Bordeaux, France) and a Generic HIV Viral Load kit (Biocentric) was used to quantify the viral load levels. The quantification method has a lower detection limit of 1,550 copies/mL, which we defined as the threshold for detectable viremia.

2. The unadjusted IRRs and the adjusted IRRs (for example, Tables 2 and S4 for men) should ideally be shown on the same table, to enable direct comparison between unadjusted and adjusted results. Also, the adjusted information is important, and deserves greater prominence than it might receive if it is solely in the supplementary materials.

Response: We agree with the Reviewer. We now include the adjusted IRRs by ART category, ART period, and year in Tables 2–3 and Table S1. In addition, we show the IRRs for the other covariates (e.g., age, marital status, etc.) in Tables S4–S6.

Reviewer #2 (Remarks to the Author):

Comments on revised manuscript, NCOMMS-19-13204A, Vandormael et al

The authors have done a thorough job in responding to earlier comments. A few additional comments on the revised version are below.

Main comments:

Results:

1. Para 1, line 4: would insert text to make it explicit that an average of 35% of all eligible enumerated adult participants in Hlabisa took part in HIV testing each year (0.85 x 0.41).

Response: As recommended by the Reviewer, we present the results of the annual eligibility, contact, and testing rates in the *Results* section and Table 1, and refer the reader to Larmarange et al. for further details. As noted in the first round of responses, it is not necessary for HIV-positive and HIV-negative participants be tested every year for HIV incidence rate estimation. At a minimum, what is needed is an earliest HIV-negative test result followed by at least one valid HIV test result. (We make this point in the revised paragraph 7 of the *Discussion* section and discuss the use of a robust statistical method to deal with the interval censoring problem). In the *Results* section, we therefore explicitly report the testing results related to the inclusion criteria for this study, which is the number of HIV-negative participants that were eligible for the incidence cohort (Column 9) and the number and percentage of these participants that had a repeat HIV test (Columns 10, 11).

2. Para 2, line 1 (and Table 1): Origin of the study group of 22,239 participants is not clear. Authors refer to another paper Larmarange in the response document (response #21) and comment that the flow/consort diagram that would explain the study group is now cited in the revised manuscript but in fact this citation is not in the Reference list. Since this study group is central to this paper, authors should briefly explain it here.

Response: Following the Reviewer's comment, we now move the text describing the origin of the 22,239 repeat-testers to the first paragraph of the *Statistical Analysis* section. We have also made this text clearer by writing:

We measured trends in the incidence of HIV infection using a prospectively followed cohort of repeat-testers. To be included in the incidence cohort, participants had to have a first HIV-negative result followed by at least one valid test result.

We repeat the description of the origin of the HIV incidence cohort in the second paragraph of the *Results* section by writing:

Of the participants that tested for HIV in the surveillance system, there were 22,239 repeat-testers who had a first negative test result followed by at least one test result.

The Larmarange et al. reference is now included in the first sentence of the *Results* section.

3. On a related note, origin of all of the values in Table 1, column 11 are not clear; it is puzzling that all of the values in column 11 are consistently greater than the numerators listed in column 9. This needs clarification.

Response: We are not sure if we understand this question: the values in Column 11 (the last column of Table 1) are percentages. Column 9 (“HIV-negative Tested”) represents the cumulative number of HIV-negative participants since 2003 that were eligible for entry into the HIV cohort. Because of 2003 and 2004, the cumulative number of eligible HIV-negative participants in 2005 is higher than the number that were contacted and tested in 2005 only (see Column 6 “HIV Tested”). To avoid possible confusion and make it more explicit, we now rename Column 9 to “HIV-negative Eligible”. Column 10 (“Repeat-testers”) shows the number of eligible HIV-negative participants that had a repeat-test, entered the HIV cohort, and contributed person-time to the incidence analysis. Column 9 is the numerator and Column 10 the denominator for the Column 11 percentages (Column 10 is smaller than Column 9).

We repeat our earlier point to Comment #3. If a participant had a first and last HIV-negative test in 2006 and 2010, then he/she contributes HIV-negative exposure time for years 2006, 2007, 2008, 2009, and 2010, even if he/she did not test in the middle years. We now reflect the above points in the Table 1 footnote, where we write:

⁵Shows the cumulative number of HIV-negative participants since 2003 that were eligible for entry into the HIV incidence cohort from 2005. ⁶Shows the number of eligible HIV-negative participants that had a repeat-test, entered the HIV cohort, and contributed person-time to the incidence analysis. For example, if a participant had a first and last HIV-negative test in 2006 and 2010, then he/she is included in the years 2006, 2007, 2008, 2009, and 2010, irrespective of the number of missed tests. The last column gives the annual percentage of eligible HIV-negative participants that entered the incidence cohort and contributed person time.

4. Para 8 on MV regression: the declines do not look monotonic to me. They are larger and more consistent for the men than for the women so they might be described as monotonic among the men but I would not use that term for the declines seen among the women. Also Model 1 has only 3 values so hard to call any pattern monotonic. Same point re thonotonic'for para 6 of Discussion

Response: These results refer to the *adjusted* IRRs for men and women, which we presented under the *Multivariate Regression Results* section. For example, in Model 1 of Table 2/Table S4, the adjusted male IRRs decrease monotonically with increasing ART coverage from 0.72 to 0.64 to 0.52. For Table 3/Table S5, the female IRRs decrease monotonically from 0.99 to 0.87 to 0.68. Following the Reviewer’s comment, we removed the word *monotonic* and renamed the section to *Adjusted IRRs*, where we write:

We show the multivariable regression results for men and women by ART coverage category, ART eligibility period, and year in Tables 2–3 and the full results in Tables S4–S5. For Model 1, declines in the *adjusted* IRRs were associated with increased opposite-sex ART coverage, holding HIV prevalence and other key risk factors for HIV acquisition constant.

5. *Suggest authors provide context on whether and how well data from one-third of Hlabisa adult population represent all of Hlabisa. And how well do data from Hlabisa represent all of SA or other regions of southern Africa, especially given rural predominance.*

Response: We are not sure where the Reviewer gets the one-third number from (possibly from Comment #1). In Table 1, we show that 80% of all eligible participants in the surveillance area tested at least once for HIV by 2017 (Column 8 “Ever Tested”), which we also report in paragraph 1 of the *Results* section. In the second and third paragraphs of the *Methods* section, we describe how field-workers visit all household in the surveillance area to undertake interviews and collect samples for population-based HIV testing. Because this sample is drawn from a total population census within a demographic health surveillance site, we accounted for non-response using inverse probability weighting to help mitigate the effect of non-participation in those who did not test during observation, as reported in paragraph 4 of the *Statistical Methods* section and updated in paragraph 8 of the *Discussion* section. Following the Reviewer’s comment, we now write in paragraph 1 of the *Methods* section that Hlabisa is typical of a rural South African setting:

Households are mostly scattered across the predominantly rural landscape with several informal peri-urban settlements and a single urban township, which is typical of a rural South African setting.

6. *Table 1, as noted above: Column 9 numerator values are consistently smaller than column 11 values—please explain in Methods and or with a table footnote.*

Response: Following our detailed response to Comment #3, we have made clarifications to the footnote of Table 1.

7. *Table 1: Would also add Average of Total row at bottom of the table since the text in the Results section refers to these average values.*

Response: Following the Reviewer's recommendation, we now add an Average row at the bottom of Table 1.

8. *Table S4 and S5: add reference group for age bands (ie ages 15-19 y). Specify if HIV prevalence refers to HIV prevalence among opposite sex.*

Response: Thank you for these comments. The age and other reference groups have been added (they were previously in the footnotes) and the opposite-sex HIV prevalence labels have been added to Tables S4–S5.

9. *Table S6: Does HIV prevalence here refer to total in Hlabisa community?*

Response: Yes, it does. To make this clearer we now write the column label in Table S6 as "Overall HIV prevalence."

Minor comments:

Methods, Statistical analysis:

10. *First two paragraphs are more about survey methods than about statistical methods.*

Response: We agree with the Reviewer. These two paragraphs have now been moved to the *Demographic and HIV survey methods* section.

11. *Why use 1550 c/mL as a cut point? This is not the usual convention and while the authors refer to their prior work (ref 22) as an explanation, a brief sentence to justify the cut point would be helpful to the reader.*

Response: The viral load level of 1,550 copies/mL is the lowest detection limit of the quantification method used by the AHRI laboratory for dried blood spots, which was standard for the field at the time of testing. We now write in paragraph 3 of the *Demographic and HIV survey methods* section:

Viral load levels have been obtained from all HIV-positive samples collected from the HIV survey in 2011, 2013, and 2014. Nucleic acid was extracted with NucliSENS EasyMag (Bordeaux, France) and a Generic HIV Viral Load kit (Biocentric) was used to quantify the viral load levels. The quantification method has a lower detection limit of 1,550 copies/mL, which we defined as the threshold for detectable viremia.

Results:

11. Para 5: Text on detectable viremia should refer to Figure 3 to illustrate the data points described. Just need to add it in parentheses at right location.

Response: Thank you for this suggestion, this has been done.

12. Para 9 on MV regression: "...and women had a significantly higher adjusted IRR..." This refers to Table S6 which does not list Women as a variable; is this sentence referring to the "not married" variable?

Response: We thank the Reviewer for pointing this out. We have now revised the variable label in Table S6 to reflect that women are the reference group for the circumcised and uncircumcised men.

13. Figure 1: Would say that incidence began to decline "after" 2012 (and after 2014 for women) rather than "from 2012" or "from 2014," as the latter suggests the decline became evident starting in 2012. This is minor semantic point but it took me some time to see the authors mean the decline is evident after 2012/2014.

Response: We have made this change in Figure 1. Where possible, we have made similar changes in the main text. In other cases, for example, we have kept the wording "between 2012 and 2017" or "from 2012 to 2017" because we use the point estimates from 2012 and 2017 to calculate the percentage reduction in HIV incidence.

14. Fig 4: would add 95% CIs, as in Figure 1

Response: We have added 95% CIs to Figure 4.

REVIEWERS' COMMENTS:

Reviewer #1 (Remarks to the Author):

None

Reviewer #2 (Remarks to the Author):

The authors have responded to all comments in a satisfactory manner.

I have only two minor suggestions:

1) re viral load lower limit of detection (LOD) of 1550 c/mL:

The test kit description on the Biocentric website uses the following name for the assay, "Generic HIV Charge Virale" even on the english website, rather than "Generic HIV Viral Load" kit. See <https://www.biocentric.com/copie-de-generic-hiv-charge-virale?lang=en>

According to Biocentric, the kit has a lower limit of detection of 390 c/mL. In the paper the authors clearly refer to the lower LOD for the method they used as 1550 and refer to a paper that is clearly about DBS. Nonetheless, the authors might want to make it easy for the reader by adding "DBS" to the phrasing and using name of kit as indicated on Biocentric website, eg: "Viral load levels were obtained from all HIV-positive DBS samples Nucleic acid was extracted... and Generic HIV Charge Virale (Biocentric, Bandol France)..."

2) Results: Para 9: revised text says "...and uncircumcised/circumcised men had a significantly lower adjusted IRR..." Suggest revising to "...and both uncircumcised and circumcised men had a significantly lower adjusted IRR..."

REVIEWERS' COMMENTS:

Reviewer #2 (Remarks to the Author):

The authors have responded to all comments in a satisfactory manner.

I have only two minor suggestions:

1) re viral load lower limit of detection (LOD) of 1550 c/mL:

The test kit description on the Biocentric website uses the following name for the assay, "Generic HIV Charge Virale" even on the english website, rather than "Generic HIV Viral Load" kit. See

<https://www.biocentric.com/copie-de-generic-hiv-charge-virale?lang=en>

According to Biocentric, the kit has a lower limit of detection of 390 c/mL. In the paper the authors clearly refer to the lower LOD for the method they used as 1550 and refer to a paper that is clearly about DBS. Nonetheless, the authors might want to make it easy for the reader by adding "DBS" to the phrasing and using name of kit as indicated on Biocentric website, eg: "Viral load levels were obtained from all HIV-positive DBS samples Nucleic acid was extracted... and Generic HIV Charge Virale (Biocentric, Bandol France)..."

Response: Following the reviewer's comment, we now write:

From the HIV survey in 2011, 2013, and 2014, we collected viral load measurements from all HIV-positive DBS samples. Nucleic acid was extracted from the DBS samples with NucliSENS EasyMag (Bordeaux, France) and a Generic HIV Charge Virale (Biocentric, Bandol, France) test was used to quantify the viral load levels.

2) Results: Para 9: revised text says "...and uncircumcised/circumcised men had a significantly lower adjusted IRR..." Suggest revising to "...and both uncircumcised and circumcised men had a significantly lower adjusted IRR..."

Response: We agree and have made this edit.